# HIDE & SEEK: TRANSFORMER SYMMETRIES OBSCURE SHARPNESS & RIEMANNIAN GEOMETRY FINDS IT

## ABSTRACT

The concept of sharpness has been successfully applied to traditional architectures like MLPs and CNNs to predict their generalization. For transformers, however, recent work reported weak correlation between flatness and generalization. We argue that existing sharpness measures fail for transformers, because they have much richer symmetries in their attention mechanism that induce directions in parameter space along which the network or its loss remain identical. We posit that sharpness must account fully for these symmetries, and thus we redefine it on a quotient manifold that results from quotienting out the transformer symmetries, thereby removing their ambiguities. Leveraging tools from Riemannian geometry, we propose a fully general notion of sharpness, in terms of a geodesic ball on the symmetry-corrected quotient manifold. In practise, we need to resort to approximating the geodesics. Doing so up to first order yields existing adaptive sharpness measures, and we demonstrate that including higher-order terms is crucial to recover correlation with generalization. We present results on diagonal networks with synthetic data, and show that our geodesic sharpness reveals strong correlation for real-world transformers on ImageNet.

## 1 INTRODUCTION

Predicting generalization performance of neural networks, i.e., the difference between performance on training data and that on a held-out test set, is an open problem. Metrics predictive of generalization performance are useful because, for example, one can explicitly regularize such a metric during training to improve generalization (as in Foret et al. (2021)), or to study generalization more broadly.

There is a long history of hypotheses relating sharpness and generalization, with conflicting theories and conflicting evidence (Hochreiter & Schmidhuber, 1994; Andriushchenko et al., 2023). Generalization has been speculated as correlating with sharpness, it has been speculated as correlating with flatness, and recent evidence has indicated that it has little to no correlation whatsoever, especially in the case of transformers. Measures of sharpness have ranged from the trace of the Hessian to worst-case loss in a neighbourhood, and have included adaptive and relative variations (Kwon et al., 2021; Petzka et al., 2021). We wonder whether some of the confusion has resulted from the specificity of the problem these measures have sought to overcome: the issue of parameter rescalings.

In contrast, we believe rescaling (Dinh et al., 2017) is a special case of a more pervasive and general obstacle to measuring sharpness accurately, which we address here at a fundamental level and from a principled perspective: the issue of full and continuous parameter symmetries. While the relationship between sharpness and generalization is likely still a complex one, a crucial step towards studying it must be to ensure we are studying the right quantity, and to do this we must overcome the obstacle of symmetry.

In addition to discrete permutation symmetries over the parameters, continuous symmetries within the parameter space occur naturally in neural networks (NNs), and are an intrinsic, data-independent property. They emerge from the standard components we stack into larger architectures: normalization layers (Ioffe & Szegedy, 2015; Ba et al., 2016; Wu & He, 2018) induce scale invariance on the pre-normalization weights (Salimans & Kingma, 2016); homogeneous activation functions like ReLU introduce re-scaling symmetries between pre- and post-activation weights (Dinh et al., 2017); some normalization layers and softmax impose translation symmetries in the preceding layer's

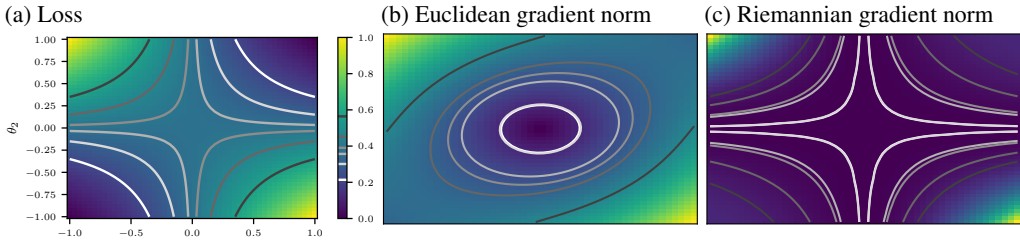

(a) Loss          (b) Euclidean gradient norm          (c) Riemannian gradient norm

Figure 1: **Quantities from the Riemannian quotient manifold respect the loss landscape's symmetry; Euclidean quantities do not.** We illustrate this here for a synthetic least squares regression task with a two-layer NN, where $x \mapsto \theta_2 \theta_1 x$ with scalar parameters $\boldsymbol{\theta} \in \mathbb{R}^2$ and input $x \in \mathbb{R}$ (i.e. each layer is a linear function). The NN is re-scale invariant, i.e. has $\mathrm{GL}(1)$ symmetry: For any $\alpha \in \mathbb{R} \setminus \{0\}$, the parameters $(\theta_1', \theta_2') = (\alpha^{-1}\theta_1, \alpha\theta_2)$ represent the same function. (a) The loss function inherits this symmetry and has hyperbolic level sets. (b) The Euclidean gradient norm does not share the loss function's geometry and changes throughout an orbit where the NN function remains constant. (c) The Riemannian gradient norm follows the loss function's symmetry and remains constant throughout an orbit, i.e., it does not suffer from ambiguities for two points in parameter space that represent the same NN function.

biases (Kunin et al., 2021). Arguably, almost any NN, along with its corresponding loss, exhibit symmetries and can therefore represent the *same* function using *different* parameter values (Figure 1a).

Adaptive flatness accounts for some symmetries, both element-wise and filter-wise, but does not capture the *full* symmetry of the attention mechanism, represented by $\mathrm{GL}(h)$ (re-scaling by invertible $h \times h$ matrices, where $h$ is the hidden dimension), as we will discuss later. We argue that flatness measures for generalization should be constructed in a symmetry-invariant fashion, such that parameters which are equivalent w.r.t. the neural net function are treated identically.

Aiming to break the cycle between discovery of a symmetry and techniques to deal with it, we ask:

*Can we provide a one-fits-many recipe to develop symmetry-invariant quantities for a wider range of symmetries?*

We answer this question positively in this paper by proposing a principled approach to eliminate ambiguities stemming from symmetry. Essentially, this boils down to using a geometry that correctly captures the symmetry-imposed equivalence of parameters. We apply concepts from Riemannian geometry to work on the Riemannian quotient manifold implied by a symmetry group (Boumal, 2023, Chapter 9). We thus identify objects on the quotient manifold—like the Riemannian metric and gradient—and show how to translate them back to the Euclidean space.

Our contributions are the following :

(a) We introduce the application of Riemannian geometry (Boumal, 2023) to the study of neural network parameter space symmetry: we propose using geometry from the quotient manifold induced by a symmetry as a general recipe to remove symmetry-induced ambiguities in parameter space. We do so by translating concepts like gradients from the quotient manifold back to the original space through horizontal lifts.

(b) We thus propose *geodesic sharpness*, a novel adaptive sharpness measure. Through Taylor expansions in our refined geometry, we show that (a) symmetries do introduce curvature into the parameter space, and (b) ignoring that curvature results in previous adaptive sharpness measures.

(c) We solve *geodesic sharpness* analytically for diagonal networks, where we find a strong correlation between sharpness and generalization. We apply our approach to the $\mathrm{GL}(h)$ symmetry in the attention mechanism, which has been unstudied and is higher-dimensional than previously considered symmetries. We empirically verify our approach on large vision transformers and find a stronger correlation than any previously seen in the literature (that we are aware of) between sharpness and generalization, both for in-distribution and for all distributional shifts we tested.

## 2 RELATED WORK

**Symmetry versus reparameterization**   Recently, Kristiadi et al. (2023) pointed out how to fix ambiguities stemming from incorrect reparameterization, i.e. a change of variables to a *new* parameter space. They show that invariance under reparameterization follows by correctly transforming the (often implicitly treated) Riemannian metric, into the new coordinate system. Our work focuses on invariance of the parameter space $\overline{\mathcal{M}}$ under a symmetry group $\mathcal{G}$ with action $\boldsymbol{\psi} : \mathcal{G} \times \overline{\mathcal{M}} \rightarrow \overline{\mathcal{M}}, \ (g, \boldsymbol{\theta}) \mapsto \boldsymbol{\psi}(g, \boldsymbol{\theta})$ that operates on a *single* parameter space.

**Symmetry teleportation:**   Other ways to circumvent the ambiguity is to view it as a degree of freedom and develop adaptation heuristics to improve algorithms which are not symmetry-agnostic (Zhao et al., 2022a).

**Geometric constraints & NN dynamics**   Some works aim at understanding the footprint of symmetries onto the parameter space, e.g. in terms of geometrical constraints on derivatives like the gradient or Hessian, or conserved quantities preserved throughout training (Kunin et al., 2021). We choose a different approach in this work and aim at eliminating the ambiguity stemming from a symmetry in a principled manner. Our approach is to take the parameter space's quotient w.r.t. the underlying symmetry group. While it reduces to many of the proposed post-hoc 'fixes' for simpler, well-studied, symmetries like GL(1), we illustrate this approach for larger symmetry group that is prevalent in the attention mechanism of modern neural network architectures. This symmetry is GL(h) with h denoting the attention head's dimension, and removes more degrees of freedom than previously studied symmetries.

Kunin et al. (2021) study the impact of continuous differentiable parameter space symmetries on the Euclidean gradient, Hessian, and gradient flow dynamics. They show that such symmetries lead to geometrical constraints on the Euclidean derivatives, which themselves lead to conserved quantities throughout gradient flow training. They focus on one-dimensional symmetries, such as translation, scale, and re-scale symmetry, whose groups are isomorphic to $\mathbb{R}$ or $\mathbb{R}^+$. Their approach is based on augmenting a symmetric function $\overline{f} : \overline{\mathcal{M}} \rightarrow \mathbb{R}$ to $\overline{F} = \overline{f} \circ \boldsymbol{\psi} : \overline{\mathcal{M}} \times \mathcal{G} \rightarrow \mathbb{R}$ with $\overline{F}\big|_g = \overline{f} \ \forall \boldsymbol{\theta} \in \overline{\mathcal{M}}$. The geometric constraints follow from differentiation in the 'augmented space', which leads to interactions between the parameter space $\overline{\mathcal{M}}$ and the symmetry group $\mathcal{G}$.

This is different to our approach which tackles the reverse direction. We consider the geometry on the quotient space induced by a symmetry group. This is akin to restricting the function $f\big|_{\overline{\mathcal{M}}/\mathcal{G}} : \mathcal{M} \rightarrow \mathbb{R}$ to the quotient space rather than an augmented space. We then 'lift' objects that reside in the quotient space and feature its more sophisticated, symmetry-aware geometry, back into the original space where computations happen. This should be seen as a form of symmetry correction to the Euclidean objects, e.g. the gradient or Hessian.

Our approach can also be applied to the one-dimensional symmetries studied by Kunin et al. (2021). We go beyond those cases by considering a higher-dimensional variant of re-scale symmetry, represented by the group of invertible matrices GL. To the best of our knowledge, this symmetry has not been studied in the context of neural networks, although it is present in the attention mechanism of large language models. We believe that the principled approach provided by quotient manifolds enables a general treatment of symmetries, and allows going beyond one-dimensional cases that lead to a more aggressive dimensionality reduction.

## 3 PRELIMINARIES: DEFINITIONS, NOTATION, AND MATHEMATICS

**Generalization measures:**   We consider a neural network $f_{\boldsymbol{w}}$ with parameters $\boldsymbol{w} \in \mathbb{R}^D$ that is trained on a data set $\mathbb{D}_{\text{train}}$ using a loss function $\ell$ by minimizing the empirical risk

$$L_{\mathbb{D}_{\text{train}}}(\boldsymbol{w}) \coloneqq \frac{1}{|\mathbb{D}_{\text{train}}|} \sum_{(\boldsymbol{x}, \boldsymbol{y}) \in \mathbb{D}} \ell(f_{\boldsymbol{w}}(\boldsymbol{x}), \boldsymbol{y})$$

Our goal is to compute a quantity on the training data that is predictive of the network's generalization, i.e. performance on a held-out data set.

**Sharpness**   One avenue to assess generalization is through the concept of sharpness, i.e.,how much the loss changes when weights are perturbed. One can do this by considering expected sharpness or worst-case sharpness over a neighbourhood of parameters, $S_{\text{avg}}$ and $S_{\text{max}}$, respectively. Usually,

$$S_{\text{avg}} = \mathbb{E}_{\mathbb{S}\sim\mathbb{D}}\left[L_S(w + \delta) - L_S(w)\right], \quad \delta \sim \mathcal{N}(0, \rho^2) \tag{1}$$

$$S_{\text{max}} = \mathbb{E}_{\mathbb{S}\sim\mathbb{D}}\left[\max_{\|\delta\|_2\leq\rho}\left(L_S(w + \delta) - L_S(w)\right)\right] \tag{2}$$

where $\mathbb{S} \subset \mathbb{D}_{\text{train}}$, $|\mathbb{S}| = m$. Near critical points, these measures are closely related to the Hessian (and thus the curvature of parameter space): $S_{\text{avg}} \sim \text{Tr}(H)$ and $S_{\text{max}} \sim \lambda_{\text{max}}(H)$.

**Problems with sharpness measures under symmetries:**   A problem with the above Hessian-based sharpness measures is that they can assume different values at points in parameter space where the neural network represents the same function. This is the case whenever the network has symmetries, and is therefore very common for almost any architecture, e.g. scale/rescale symmetries.

**Adaptive sharpness:**   To fix the above inconsistency, Kwon et al. (2021) proposed adaptive sharpness (invariant under special symmetries), and Andriushchenko et al. (2023) utilize adaptive notions of sharpness, which can be shown to be invariant to element-wise scalings :

$$S_{\text{max}}^{\rho}(w, c) = \mathbb{E}_{\mathbb{S}\sim\mathbb{D}}\left[\max_{\|\boldsymbol{\delta}\odot\boldsymbol{c}^{-1}\|_p\leq\rho} L_{\mathbb{S}}(w + \delta) - L_{\mathbb{S}}(w)\right] \tag{3}$$

where $\mathbb{S}$ is a batch of size $m$ drawn from the training data, and $\boldsymbol{c}$ a vector with respect to which the adaptive sharpness is considered, usually taken to be $|\boldsymbol{w}|$ Kwon et al. (2021).

**The problem:**   Adaptive sharpness only considers a special symmetry. But symmetries of transformers go beyond the invariance that adaptive sharpness was built for. Maybe unsurprisingly, Andriushchenko et al. (2023) find inconsistent trends for adaptive sharpness in transformers versus other architectures. We hypothesize this is related to adaptive sharpness not accounting for the full symmetry in transformers. In this paper, we aim to fix this. The central question is: *If adaptive sharpness is the fix for a special symmetry, can we do something similar for the symmetries of transformers to fix the above inconsistency?*

### 3.1   Symmetries in Neural Networks

Here, we give a brief overview and make more concrete the notion of NN symmetries, focusing on those previously studied in Kunin et al. (2021).

Those symmetries lead to rather small effective dimensionality reduction as they are often of $\text{GL}(1)$ or $\text{GL}^+(1)$, but they can still impact the network behaviour considerably. Let $\boldsymbol{\theta}$ denote the parameters of a neural network, $\mathbf{1}_{\mathcal{A}}$ a binary mask, and $\mathbf{1}_{\neg\mathcal{A}}$ its complement such that their sum is a vector of ones, $\mathbf{1}_{\mathcal{A}} + \mathbf{1}_{\neg\mathcal{A}} = \mathbf{1}$. Let $\boldsymbol{\theta}_{\mathcal{A}} := \boldsymbol{\theta} \odot \mathbf{1}_{\mathcal{A}}$ with $\odot$ the element-wise product. Further, let $\mathcal{A}_{1,2}$ be two disjoint subsets, $\mathcal{A}_1 \cap \mathcal{A}_2 = \emptyset$ with masks $\mathbf{1}_{\mathcal{A}_1}, \mathbf{1}_{\mathcal{A}_2}$. Then we have the following common symmetries, characterized by their symmetry group $\mathcal{G}$, such that for any $g \in \mathcal{G}$ the parameter $\psi(g, \boldsymbol{\theta})$ represents the same function as $\boldsymbol{\theta}$:

- **Translation:** $\psi(\boldsymbol{\alpha}, \boldsymbol{\theta}) = \mathbf{1}_{\mathcal{A}} \odot \boldsymbol{\alpha} + \boldsymbol{\theta}$ with $\boldsymbol{\alpha} \in \mathbb{R}^h$
- **Scaling:** $\psi(\alpha, \boldsymbol{\theta}) = \alpha\boldsymbol{\theta}_{\mathcal{A}} + \boldsymbol{\theta}_{\neg\mathcal{A}}$ with $\alpha \in \mathbb{R}_{>0}$
- **Re-scaling:** $\psi(\alpha, \boldsymbol{\theta}) = \alpha\boldsymbol{\theta}_{\mathcal{A}_1} + {}^1\!/{}_\alpha\boldsymbol{\theta}_{\mathcal{A}_2} + \boldsymbol{\theta}_{\neg(\mathcal{A}_1\vee\mathcal{A}_2)}$ with $\alpha \in \mathbb{R}_{>0}$

Their associated groups are $\mathcal{G} = \mathbb{R}^h, \text{GL}^+(1), \text{GL}^+(1)$, respectively. In practise, there may be multiple symmetries acting onto disjoint parts of the parameter space. Note that the re-scaling symmetry is essentially the symmetry that adaptive sharpness corrects for.

### 3.2   Re-scale Symmetry of Transformers

Transformers exhibit a higher-dimensional symmetry than the previous examples; we formalize the treatment of this symmetry in the following canonical form.

**Definition 3.1** (Functional building block with GL symmetry). Consider a function $\overline{\overline{f}}(\boldsymbol{G}, \boldsymbol{H})$ on $\mathbb{R}^{m\times h} \times \mathbb{R}^{n\times h}$ that consumes two matrices $\boldsymbol{G} \in \mathbb{R}^{n\times h}, \boldsymbol{H} \in \mathbb{R}^{m\times h}$ but only uses the product

$GH^\top$, i.e. $\overline{\overline{f}}(G, H) = g(GH^\top)$ for some $g$ over $\mathbb{R}^{m \times n}$. $\overline{\overline{f}}$ is symmetric under the *general linear group*

$$\mathrm{GL}(h) \coloneqq \left\{ A \in \mathbb{R}^{h \times h} \mid A \text{ invertible} \right\} \tag{4a}$$

with dimension $\dim(\mathrm{GL}(h)) = h^2$ and action

$$\psi(A, (G, H)) = (GA^{-1}, HA^\top). \tag{4b}$$

In other words, we can insert then absorb the identity $A^{-1}A$ into $G, H$ to obtain equivalent parameters $GA^{-1}, HA^\top$ that represent the same function.

Example A.2 illustrates GL symmetry for a shallow linear net. Indeed, many popular NN building blocks feature this form—most prominently the attention mechanism in transformers—and introduce GL symmetries into the loss landscape. We give the attention symmetry in Example A.1, and we provide the symmetry for for low-rank adapters (Hu et al., 2022) in Example A.3.

Examples A.1 to A.3 are NN building blocks that introduce GL symmetries into a loss function produced by an architecture and can all be treated through the canonical form in Definition 3.1. In contrast to symmetries from Section 3.1, they lead to more drastic dimensionality reduction. Consider for example a single self-attention layer where $d = d_\mathrm{v} = d_\mathrm{k}$. The number of trainable parameters is $4d^2$ and the two $\mathrm{GL}(d)$ symmetries reduce the effective dimension to $4d^2 - 2\dim(\mathrm{GL}(d)) = 2d^2$, i.e. they render *half* the parameter space redundant. We hypothesize that the range of objects like the Euclidean Hessian's trace (Dinh et al., 2017) in the presence of a low-dimensional symmetry may be amplified for such higher-dimensional symmetries.

### 3.3 MATHEMATICAL CONCEPTS FOR RIEMANNIAN GEOMETRY

We now outline properties of manifolds that are needed for the full development of our approach. We list essential concepts here, and provide definitions and a brief review of these concepts in Appendix B. For further information, the interested reader is referred to, e.g. Lee (2003)

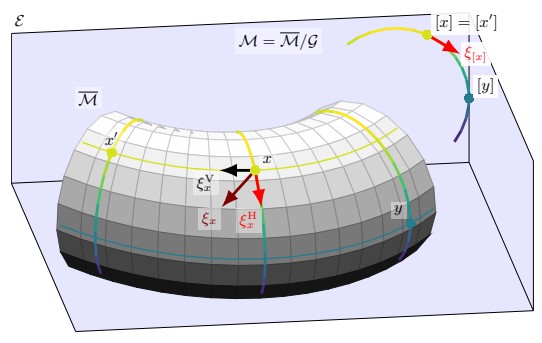

| | |
|---|---|
| $\mathcal{E}$ | Linear embedding space |
| $\overline{\mathcal{M}}$ | Total space |
| $\mathcal{M}$ | Quotient space |
| $\mathcal{G}$ | Symmetry group |
| $\bar{x}, \bar{y}$ | Points on the total space |
| $x, y$ | Points on the quotient space |
| $\bar{\xi}_{\bar{x}}$ | Tangent vector in the tangent space at point $\bar{x}$, $\mathrm{T}_{\bar{x}}\overline{\mathcal{M}}$ |
| $\xi_x$ | Tangent vector in the tangent space at point $x$, $\mathrm{T}_x\mathcal{M}$ |
| $\bar{\xi}_{\bar{x}}^\mathrm{V}$ | Vertical component of $\bar{\xi}_{\bar{x}}$ in the vertical space $\mathrm{V}_{\bar{x}}\overline{\mathcal{M}}$ |
| $\bar{\xi}_{\bar{x}}^\mathrm{H}$ | Horizontal component of $\bar{\xi}_{\bar{x}}$ in the horizontal space $\mathrm{H}_{\bar{x}}\overline{\mathcal{M}} \simeq \mathrm{T}_x\mathcal{M}$, horizontal lift of $\xi_x$ |

Figure 2: Illustrative sketch relating total and quotient space and their tangent spaces. A tangent vector at a point in total space, $\bar{\xi}_{\bar{x}} \in T_{\bar{x}}\overline{\mathcal{M}}$ can be decomposed into a horizontal component $\bar{\xi}_{\bar{x}}^\mathcal{H}$ and a vertical component $\bar{\xi}_{\bar{x}}^\mathcal{V}$. The vertical component points along the direction where the quotient space $x = [\bar{x}]$ remains unaffected. The horizontal component points along the direction that changes the equivalence class. We can use $\bar{\xi}_{\bar{x}}^\mathcal{H}$ as a representation of the tangent vector $\xi_x \in T_x\mathcal{M}$ on the quotient space. The component $\bar{\xi}_{\bar{x}}^\mathcal{H}$ represents the *horizontal lift* of $\xi_x$.

**Ambient embedding space** We assume our parameter manifold to be embedded in linear Euclidean space $\mathcal{E} \simeq \mathbb{R}^d$ with $d$ the number of parameters. We can think of $\mathcal{E}$ as the *computation space*. For instance, for a loss function $\overline{\overline{\ell}} : \mathcal{E} \to \mathbb{R}, \boldsymbol{\theta} \mapsto \overline{\overline{\ell}}(\boldsymbol{\theta})$, we can use ML libraries to evaluate its value, as well as its Euclidean gradient

$$\mathrm{grad}_{\boldsymbol{\theta}} \overline{\overline{\ell}} = \left( \frac{\partial \overline{\overline{\ell}}(\boldsymbol{\theta})}{\partial \theta_i} \right)_{i=1,\dots,d} \in \mathbb{R}^d. \tag{5}$$

Because the geometry of $\mathcal{E}$ is flat, i.e. uses the standard metric $\langle \boldsymbol{\theta}_1, \boldsymbol{\theta}_2 \rangle := \boldsymbol{\theta}_1^\top \boldsymbol{\theta}_2$, this object consists of partial derivatives. However, the Riemannian generalization will add correction terms. In what follows we consider only the restriction of objects like $\bar{\bar{\ell}}$ to the parameter space.

**Definition 3.2.** We take $\overline{\mathcal{M}}$ to be the manifold of parameters of our network, and consider it as being an embedded submanifold of an ambient vector space $\mathcal{E}$, the computational space of matrices on which all our numerical calculations are done. We call $\overline{\mathcal{M}}$ the total space. On the total space we have a loss function $\bar{\ell} : \overline{\mathcal{M}} \to \mathcal{R}$.

As discussed in the previous section, we intend on calculating derivatives/geometric quantities after removing the symmetries from our neural architecture. The symmetry relation induces natural equivalence classes, which we write $[x]$, and explain in Appendix B.1 We let $\mathcal{M} = \overline{\mathcal{M}} / \sim$ represent the *quotient* of the original parameter space manifold by the equivalence relation associated with the symmetry (Appendix B.2). We also require *tangent vectors*; these are straightforward on the total space $\overline{\mathcal{M}}$, but the tangent space of the quotient manifold, $\mathcal{M}$, in turn requires more machinery: *vertical* and *horizontal spaces*, and corresponding *lift*. These concepts are all defined in Appendix B.3.

If endow our total space $\overline{\mathcal{M}}$ with a smooth inner product over its tangent vectors then we end up with a *Riemannian manifold* (defined in Appendix B.4). This construction will let us analyze several differential objects that live on quotient manifolds, on the ambient space in natural way, as we will see. Furthermore, this allows us to define the horizontal space as the orthogonal complement of the vertical space (also in Appendix B.4), and to define a *Riemannian gradient* (Appendix B.5). Most properties from the Euclidean case are still true for the Riemannian gradient, but of particular interest to us is the fact that the direction of $\text{grad} f(x)$ is still the steepest-ascent direction of $f$ at a point $x$.

We additionally make use of the notions of *geodesic curves*. Intuitively, geodesic curves can either be seen as curves of minimal distance between two points on a manifold $\overline{\mathcal{M}}$, or equivalently, as curves through a given point with some initial velocity, and whose acceleration is zero— a generalization of Euclidean straight lines. See Appendix B.6 for details.

## 4 GEODESIC SHARPNESS

We posit that adaptive sharpness measures should take into account the geometry of the quotient parameter manifold, that arises after removing symmetries from the parameter space. We base our sharpness measure on the notion of geodesic ball: the set of points that can be reached by geodesics starting at a point $p$ and whose initial velocity has a norm smaller than $\rho$. In $\mathbb{R}$ this is just the usual definition of a ball, since the geodesics are straight lines. Putting it all together, if $\bar{\xi} \in H_w$ is a horizontal vector, and $\bar{\gamma}(t)$ is a geodesic starting at $w$ and with initial velocity $\bar{\xi}$:

$$S_{\max}^\rho(w) = \mathbb{E}_{\mathbb{S} \sim \mathbb{D}} \left[ \max_{||\bar{\xi}||_{\bar{\gamma}(0)} \leq \rho} L_S(\bar{\gamma}_{\bar{\xi}}(1)) - L_S(\bar{\gamma}_{\bar{\xi}}(0)) \right], \tag{6}$$

If the initial velocity, $\bar{\xi}$, y, is a horizontal vector, then the velocity of the geodesic, $\dot{\bar{\gamma}}_{\bar{\xi}}$, will stay horizontal. The choice of $t = 1$ in $\bar{\gamma}_{\bar{\xi}}(1)$ is not as arbitrary as it seems (do Carmo, 1992) : for a positive $a$, $\bar{\gamma}_{\bar{\xi}}(at) = \bar{\gamma}_{a\bar{\xi}}(t)$.

When we do not have an analytical solution for the geodesic, we can use the approximation:

$$\bar{\gamma}^i(t) = \bar{\gamma}^i(0) + \bar{\xi}^i t - \frac{1}{2}\Gamma^i_{kl}\bar{\xi}^k\bar{\xi}^l t^2 + \mathcal{O}(\bar{\xi}^3) \tag{7}$$

where $\bar{\xi} = (\bar{\xi}^i)$ is the initial (horizontal) velocity, and $\Gamma^i_{kl}$ are the Christoffel symbols. With this approximation Eq. 6 becomes

$$S_{\max}^\rho(w) = \mathbb{E}_{\mathbb{S} \sim \mathbb{D}} \left[ \max_{||\bar{\xi}||_{\bar{\gamma}(0)} \leq \rho} L_S \left( \bar{\gamma}_{\bar{\xi}}(0) + \bar{\xi}^i - \frac{1}{2}\Gamma^i_{kl}\bar{\xi}^k\bar{\xi}^l \right) - L_S(\bar{\gamma}_{\bar{\xi}}(0)) \right], \tag{8}$$

We show geodesic sharpness reduces to adaptive sharpness measures in Appendix F.1.

## 5 GEODESIC SHARPNESS IN PRACTICE

We now apply geodesic sharpness to concrete examples. A fully worked out scalar toy model is provided in Appendix D. Following previous works by Dziugaite et al. (2020);Kwon et al. (2021);Andriushchenko et al. (2023), we use the Kendall rank correlation coefficient (Kendall, 1938) in the empirical validations of our approach:

$$\tau(t, s) = \frac{2}{M(M-1)} \sum_{i<j} \text{sign}(t_i - t_j) \, \text{sign}(s_i - s_j)$$

where $t$ and $s$ are the vectors between which we are trying to measure correlation.

### 5.1 DIAGONAL NETWORKS

We next study *diagonal linear networks*, one of the simplest non-trivial neural networks. Diagonal networks have two parameters, $u, v$, and predict a label, $y$ given an input, $x$, via $y = x^T(u \odot v)$. We consider a linear regression problem with labels $y \in \mathbb{R}^n$ and data matrix $X \in \mathbb{R}^{n \times d}$. We take as our loss $L(u, v) = ||X(u \odot v) - y||_2^2$. Our parameter manifold $\mathcal{M}$ will be $\mathbb{R}^d \times \mathbb{R}^d$. The symmetry present in these diagonal networks is that of element-wise re-scaling: $(u, v) \to (\alpha u, \alpha^{-1} v)$, leaves $\beta = u \odot v$ invariant and hence the loss.

**Metric:** At a point $(u, v) \in \mathcal{M}$, for two tangent vectors $\eta = (\eta_u, \eta_v)$, $\nu = (\nu_u, \nu_v) \in T_{(u,v)}\mathcal{M}$, we have

$$g\left[(\eta_u, \eta_v), (\nu_u, \nu_v)\right] = \sum_{i=1}^d \frac{\eta_u^i \nu_u^i}{(u^i)^2} + \frac{\eta_v^i \nu_v^i}{(v^i)^2} \tag{9}$$

**Horizontal space:** $H_{(u,v)} = \{(\eta_u, \eta_v) \in T_{(u,v)}\mathcal{M} \mid \frac{\eta_u^i}{u^i} = \frac{\eta_v^i}{v^i} \quad \forall i \in \{1, \ldots, d\}\}$

**Geodesics:** We define $B^i = \frac{\eta_u^i}{u^i} = \frac{\eta_v^i}{v^i} \forall i \in \{1, \ldots, d\}$, so that

$$\gamma(t)^i = (u(t), v(t)) = \left(u_0^i \exp(B_i t), v_0^i \exp(B_i t)\right) \forall i \in \{1, \ldots, d\} \tag{10}$$

where $u_0^i$ and $v_0^i$ are the initial positions for our parameters, i.e., the parameters that the network actually learned.

**Geodesic sharpness:** We assume that in what follows $X^T X = Id_d$, and we denote $\beta_0 = u_0 \odot v_0, \gamma_t = \left(\exp(2B^1 t), \ldots \exp(2B^d t)\right), \beta_t = (u_t \odot v_t) = \beta_0 \odot \gamma_t, \beta_* = X^T y$. Note that $\beta_*$ is just the optimal least squares predictor when $X^T X = Id$. With this notation

$$S_{\max} = \max_{||B|| \le \rho^2} \sum_i^d \left[(\beta_0^i)^2(\gamma_t \odot \gamma_t - 1)\right] - 2(\beta_0 \odot \gamma_t - 1)^T \beta_* \tag{11}$$

At a first glance, this expression does not seem to have a simple interpretation, but we Taylor expand it to second order in $B$ (since $\rho$ is supposed to be small):

$$S_{\max} \approx \max_{||B|| \le \rho^2} 4B^T r + 4B^T D_{\beta_0, \beta_*} B \tag{12}$$

where $r = \{\beta_0^i(\beta_0^i - \beta_*^i), i = 1, \ldots, d\}$, $r' = \{(\beta_0^i - \beta_*^i), i = 1, \ldots, d\}$ and $D_{\beta_0, \beta_*} = diag(\beta_0^i(2\beta_0^i - \beta_*^i)) = diag(\beta_0^i(\beta_0^i + (r')^i))$. We separate the analysis of Eq.12 into two cases:

CASE A): $r \ne 0$ AND FIRST ORDER SUFFICES    Eq.12 becomes

$$S_{\max} = \max_{||B|| \le \rho^2} 4B^T r$$

with solution $S_{\max} = 4\rho||r||$. This is essentially the gradient norm– a useful quantity for understanding generalization (Zhao et al., 2022b).

CASE B): $r = 0$  Here we necessarily have to consider the second order terms, so that Eq.12 becomes

$$S_{\max} = \max_{||\boldsymbol{B}|| \leq \rho^2} 4\boldsymbol{B}^T \boldsymbol{D}_{\boldsymbol{\beta}_0, \boldsymbol{\beta}_*} \boldsymbol{B}$$

This has the well known solution of $S_{\max} = \rho^2 \lambda_{\max}(\boldsymbol{D}_{\boldsymbol{\beta}_0, \boldsymbol{\beta}_*}) = \rho^2 \max((\beta_0^i)^2)$. This is just $||\boldsymbol{\beta}||_\infty^2$, which is the square of what we would get by using adaptive sharpness, Eq.33, with a very carefully chosen hyper-parameter $\boldsymbol{c}$. This is a quantity that is useful when our ground-truth, $\boldsymbol{\beta}^*$ is dense.

### 5.1.1 EMPIRICAL VALIDATION

**Experimental setup:**  We emulate the setup used in Andriushchenko et al. (2023). We generate a randomly distributed data matrix $\boldsymbol{X}$, a random ground-truth vector $\boldsymbol{\beta}^*$ that is 90% sparse, and we train 50 diagonal networks to $10^{-5}$ training loss on a regression task. We take $d = 200$. We solve the maximum sharpness optimization problem for the geodesics using Lagrange multipliers and Eq. 12.

**Results:**  We see that in this particular instance, all three notions of sharpness are able to predict, to some degree, generalization. Geodesic sharpness although closely related for diagonal networks with adaptive worst-case sharpness, does slightly better. While we do not use notions of average sharpness in other results, we included it here to further illustrate the effect that different notions of sharpness have when studying generalization. The Kendall taus indicate a strong correlation.

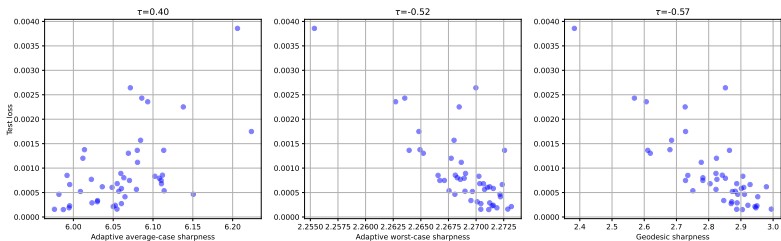

Figure 3: We show for 50 diagonal models trained on sparse data the generalization gap (here just the test loss) vs. average-case adaptive sharpness (left), worst-case adaptive sharpness (middle) and geodesic sharpness (right). The x axis is given by the various sharpness notions we consider, and the y axis is the test error. The plots for maximum sharpness are approximately linear with some dispersion.

### 5.2 ATTENTION LAYERS

We take as our computation space $\mathcal{E} := \mathbb{R}^{n \times h} \times \mathbb{R}^{m \times h} \simeq \mathbb{R}^{(n+m)h}$

In what follows we restrict our weight matrices to have full column rank.

**Assumption 5.1.** The rank of $\boldsymbol{G}, \boldsymbol{H}$ corresponds to their number of columns, $\mathrm{rank}(\boldsymbol{G}) = \mathrm{rank}(\boldsymbol{H}) = h$.

This implies $h \leq n, m$, which is usually satisfied in (multi-head) attention layers (Example A.1) for the default choices of $d_\mathrm{v}, d_\mathrm{k}$. While the weights of multi-head attention layers tend to have high rank, they are not guaranteed to be full rank. To account for this we introduce a small relaxation parameter, $\epsilon$, s.t. $\boldsymbol{G}^T \boldsymbol{G} \to \boldsymbol{G}^T \boldsymbol{G} + \epsilon I_h$. We observe that as long as $\epsilon$ is sufficiently small, it does not affect our results. In the following, we therefore restrict both $\boldsymbol{G}, \boldsymbol{H}$ to the set of fixed-rank matrices,

$$\overline{\mathcal{M}} \leftarrow \mathbb{R}_h^{n \times h} \times \mathbb{R}_h^{m \times h} \tag{13a}$$

where

$$\mathbb{R}_k^{n \times h} := \left\{ \boldsymbol{B} \in \mathbb{R}^{n \times h} \mid \mathrm{rank}(\boldsymbol{B}) = k \right\}. \tag{13b}$$

We can represent a point $\bar{x} \in \overline{\mathcal{M}}$ by a matrix tuple $(\boldsymbol{G}, \boldsymbol{H}) \in \mathbb{R}_h^{n \times h} \times \mathbb{R}_h^{m \times h}$. Its tangent space $\mathrm{T}_{\bar{x}} \overline{\mathcal{M}}$ is

$$\mathrm{T}_x \overline{\mathcal{M}} = \left\{ \eta \in \mathbb{R}^{n \times h} \times \mathbb{R}^{m \times h} \right\} \tag{14}$$

and a tangent vector $\eta \in \mathrm{T}_{\bar{x}} \overline{\mathcal{M}}$ is represented by a matrix tuple $(\eta_{\boldsymbol{G}}, \eta_{\boldsymbol{H}}) \in \mathbb{R}^{n \times h} \times \mathbb{R}^{m \times h}$.

**Metric:** We endow $\overline{\mathcal{M}}$ with the metric $\langle \cdot, \cdot \rangle_{\bar{x}} : T_{\bar{x}}\overline{\mathcal{M}} \times T_{\bar{x}}\overline{\mathcal{M}} \to \mathbb{R}$,

$$\langle \bar{\eta}, \bar{\zeta} \rangle_x = \mathrm{Tr}\left( (\boldsymbol{G}^\top \boldsymbol{G})^{-1} \bar{\eta}_{\boldsymbol{G}}^\top \bar{\zeta}_{\boldsymbol{G}} + (\boldsymbol{H}^\top \boldsymbol{H})^{-1} \bar{\eta}_{\boldsymbol{H}}^\top \bar{\zeta}_{\boldsymbol{H}} \right), \tag{15}$$

different from the Euclidean metric that simply flattens and concatenates the matrix tuples into vectors and takes their dot product, $\langle \eta, \zeta \rangle = \mathrm{Tr}\left( \eta_{\boldsymbol{G}}^\top \zeta_{\boldsymbol{G}} + \eta_{\boldsymbol{H}}^\top \zeta_{\boldsymbol{H}} \right)$. Importantly, this metric is invariant under the symmetries of the attention mechanism, and thus defines a valid metric on the quotient manifold (Absil et al., 2008) .

**Horizontal space:** $H_{\bar{x}}\overline{\mathcal{M}} = \left\{ (\bar{\xi}_{\boldsymbol{G}} + \boldsymbol{G}\boldsymbol{\Lambda}, \bar{\xi}_{\boldsymbol{H}} - \boldsymbol{H}\boldsymbol{\Lambda}^\top) \mid \xi \in T_x\overline{\mathcal{M}} \right\}$

$\boldsymbol{\Lambda}$ is the solution of the Sylvester equation $\boldsymbol{A}\boldsymbol{\Lambda} + \boldsymbol{\Lambda}\boldsymbol{A}^\top = \boldsymbol{B}$, with $\boldsymbol{A} = \boldsymbol{G}^\top \boldsymbol{G}\boldsymbol{H}^\top \boldsymbol{H}$, $\boldsymbol{B} = \boldsymbol{G}^\top \boldsymbol{G}\boldsymbol{H}^\top \bar{\xi}_{\boldsymbol{H}} - \bar{\xi}_{\boldsymbol{G}}^\top \boldsymbol{G}\boldsymbol{H}^\top \boldsymbol{H}$.

**Geodesics:** As far as we are aware there is no analytical solution for the geodesics of metric (15), so we use the approximation given by Eq.7.

For horizontal tangent vectors $(\bar{\xi}_{\boldsymbol{G}}, \bar{\xi}_{\boldsymbol{H}})$

$$\Gamma_{kl}^i \bar{\xi}_{\boldsymbol{G}}^k \bar{\xi}_{\boldsymbol{G}}^l = -\bar{\xi}_{\boldsymbol{G}}(\boldsymbol{G}^T \boldsymbol{G})^{-1}\left[ \bar{\xi}_{\boldsymbol{G}}^T \boldsymbol{G} + \boldsymbol{G}^T \bar{\xi}_{\boldsymbol{G}} \right] + \boldsymbol{G}(\boldsymbol{G}^T \boldsymbol{G})^{-1} \bar{\xi}_{\boldsymbol{G}}^T \bar{\xi}_{\boldsymbol{G}} \tag{16}$$

Proof in Appendix I.2.

### 5.3 Transformers

Transformers will have a mix of attention layers and layers with more restricted symmetries for which adaptive sharpness is more appropriate. We present in Appendix C.1 more details on how we treat the multi-layer schema of transformers. In Appendix C.2 we present Algorithm 1, which we use to solve for geodesic sharpness.

#### 5.3.1 Empirical validation: ImageNet and Vision Transformers

**Experimental setup:** We follow the lead of Andriushchenko et al. (2023), and look at models obtained from CLIP fine-tuning on ImageNet-1k from CLIP Radford et al. (2021). To be more specific, we study the already trained classifiers obtained by Wortsman et al. (2022), after fine-tuning a CLIP ViT-B/32 on ImageNet with randomly selected hyperparameters. We evaluate our measure and adaptive worst-case sharpness on the same 2048 data points from the training set of ImageNet, divided into batches of 256 points. We calculate sharpness on each batch separately and average the results. We take the generalization gap to be the difference between test and training error. We present our results also on the distribution shifts Imagenet-R (Hendrycks et al., 2021a), ImageNet-Sketch (Wang et al., 2019), and ImageNet-A (Hendrycks et al., 2021b).

**Results:** We show our results in Figure 5.3.1. We find a consistent, strong, correlation between geodesic sharpness and the generalization gap on ImageNet. This correlation is stronger than that observed with adaptive sharpness alone, and is consistently negative, implying that the geodesically sharpest models studied on ImageNet are those that generalize best.

#### 5.3.2 Empirical validation: MNLI and BERT

**Experimental Setup** We also validate our measure on 35 models from (McCoy et al., 2020), obtained after fine-tuning BERT on MNLI (Williams et al., 2018). We evaluate our measure and adaptive worst-case sharpness on the same 1024 data points from the MNLI training set, divided into batches of 128 points. As was done for ImageNet, we calculate sharpness on each batch separately and average the results.

**Results** We show our results in Figure 8. We find a consistent, strong, correlation between geodesic sharpness and the generalisation gap on MNLI. This correlation is stronger than that observed with adaptive sharpness alone, and is now positive, implying that the geodesically flattest models studied on MNLI are those that generalise best.

## 6 Limitations and future work

**Limitations:** First, while our *geodesic sharpness* is more general than previous measures, there are still symmetries for which taking the quotient may be computationally expensive or intractable. Still,

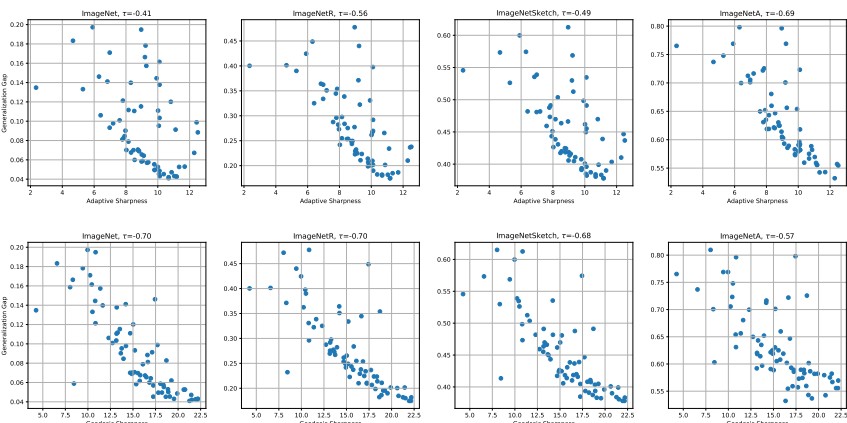

Figure 4: We show for the 72 models from Wortsman et al. (2022) the generalization gap on ImageNet (and distributional shifts) vs. worst-case adaptive sharpness (top) and geodesic sharpness (bottom).

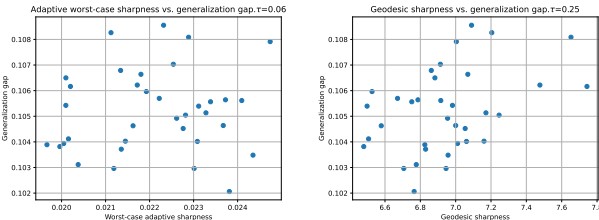

Figure 5: The generalization gap on the MNLI dev matched set (Williams et al., 2018) vs. worst-case adaptive sharpness (left) and geodesic sharpness (right), is shown for 35 models from (McCoy et al., 2020)

it provides an important tool for analysing NNs: accounting for some symmetry is better than none, and under computational constraints it could be useful as an occasional diagnostic "probe".

Our proposed sharpness measure for transformers can struggle when attention weights approach being singular. We mitigate this with a relaxation parameter. This comes at the expense of an additional hyper-parameter, though in practise we found our results were robust to this parameter.

**Future work**   The role of data and how it can be integrated into our overall framework has not been explored. A more complete understanding of the simultaneous dual invariance induced by data and parameter symmetries is sure to be invaluable in further understanding the connection of sharpness with generalisation.

As demonstrated by Foret et al. (2021), having a relatively simple quantity accessible during training that correlates to generalisation is useful, and further work might enable the creation of optimizers well-suited to large scale transformers.

## 7   CONCLUSION

In this paper, motivated by the success of adaptive sharpness measures in the study of generalization, we propose a novel adaptive sharpness measure: geodesic sharpness. We frame it in the context of Riemannian geometry and provide a one-size-fits-all recipe for including various parameter symmetries in the calculation of sharpness. We find that the symmetries introduce curvature into the parameter space and that by ignoring that curvature we recover traditional adaptive sharpness measures. We analytically investigate our measure on widely studied diagonal networks and empirically verify our approach on large scale transformers, finding a strong correlation between geodesic sharpness and generalization.

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

# Hide & Seek: Transformer Symmetries Obscure Sharpness & Riemannian Geometry Finds It  (Supplemental Material)

We provide in Table 1 a summary of correlation coefficients between sharpness and generalization for our experiments.

| | Rank correlation coefficient $\tau$ | |
|---|---|---|
| Setting | Adaptive sharpness | Geodesic sharpness |
| Diagonal networks | -0.52 | **-0.57** |
| ImageNet | -0.41 | **-0.7** |
| MNLI | 0.06 | **0.25** |

Table 1: Summary of the correlation between sharpness measures and generalization. We boldface the best performing metric

In the sections that follow, we provide additional details to supplement the main text.

## A  ADDITIONAL EXAMPLES OF GL SYMMETRIES SYMMETRIES IN NEURAL NETWORKS

**Example A.1** (Self-attention( Vaswani et al. (2017))). Given a sequence $\boldsymbol{X} \in \mathbb{R}^{t \times d}$ with $t$ tokens and model dimension $d$, self-attention (SA) uses four matrices $\boldsymbol{W}_{\mathrm{q}}, \boldsymbol{W}_{\mathrm{k}} \in \mathbb{R}^{d \times d_{\mathrm{k}}}, \boldsymbol{W}_{\mathrm{v}}, \boldsymbol{W}_{\mathrm{o}}^{\top} \in \mathbb{R}^{d \times d_{\mathrm{v}}}$ (usually, $d = d_{\mathrm{v}} = d_{\mathrm{k}}$) to produce a new $t \times d$ sequence

$$
\begin{aligned}
&\mathrm{SA}(\boldsymbol{W}_{\mathrm{q}}, \boldsymbol{W}_{\mathrm{k}}, \boldsymbol{W}_{\mathrm{v}}, \boldsymbol{W}_{\mathrm{o}}) \\
&= \mathrm{softmax}\left( \frac{\boldsymbol{X}\boldsymbol{W}_{\mathrm{q}}\boldsymbol{W}_{\mathrm{k}}^{\top}\boldsymbol{X}^{\top}}{\sqrt{d_{\mathrm{k}}}} \right) \boldsymbol{X}\boldsymbol{W}_{\mathrm{v}}\boldsymbol{W}_{\mathrm{o}}.
\end{aligned}
\tag{17}
$$

This block contains two GL symmetries: one of dimension $d_{\mathrm{k}}$ between the key and query projection weights, $\boldsymbol{G}, \boldsymbol{H} \leftarrow \boldsymbol{W}_{\mathrm{q}}, \boldsymbol{W}_{\mathrm{k}}$, and one of dimension $d_{\mathrm{v}}$ between the value and out projection weights, $\boldsymbol{G}, \boldsymbol{H} \leftarrow \boldsymbol{W}_{\mathrm{v}}, \boldsymbol{W}_{\mathrm{o}}^{\top}$. Similar to Eq. 18, we can account for biases in the key, query, and value projections by appending them to their weight,

$$
\boldsymbol{G}, \boldsymbol{H} \leftarrow \begin{pmatrix} \boldsymbol{W}_{\mathrm{k}} \\ \boldsymbol{b}_{\mathrm{k}}^{\top} \end{pmatrix}, \begin{pmatrix} \boldsymbol{W}_{\mathrm{q}} \\ \boldsymbol{b}_{\mathrm{q}} \end{pmatrix}^{\top}, \quad \boldsymbol{G}, \boldsymbol{H} \leftarrow \begin{pmatrix} \boldsymbol{W}_{\mathrm{v}} \\ \boldsymbol{b}_{\mathrm{v}} \end{pmatrix}, \boldsymbol{W}_{\mathrm{o}}^{\top}.
$$

Commonly, $H$ attention heads $\{\boldsymbol{W}_{\mathrm{q}}^{i}, \boldsymbol{W}_{\mathrm{k}}^{i}, \boldsymbol{W}_{\mathrm{v},i}^{i}, \boldsymbol{W}_{\mathrm{o}}^{i}\}_{i=1}^{H}$ independently process $\boldsymbol{X}$ and concatenate their results into the final output (usually $d_{\mathrm{k}} = d_{\mathrm{v}} = {}^{d}/{}_{H}$). This introduces $2H$ GL symmetries. Everything also applies to general attention where, instead of $\boldsymbol{X}$, independent data is fed as keys, queries, and values to Eq. 17.

**Example A.2** (Shallow linear net). Consider a two-layer linear net $\mathrm{NN}(\boldsymbol{W}_2, \boldsymbol{W}_1) = \boldsymbol{W}_2\boldsymbol{W}_1\boldsymbol{x}$ with weight matrices $\boldsymbol{W}_1 \in \mathbb{R}^{h \times d_{\mathrm{in}}}, \boldsymbol{W}_2 \in \mathbb{R}^{d_{\mathrm{out}} \times h}$ and some input $\boldsymbol{x} \in \mathbb{R}^{d_{\mathrm{in}}}$. This net has GL symmetry with correspondence $\boldsymbol{G}, \boldsymbol{H} \leftarrow \boldsymbol{W}_2, \boldsymbol{W}_1^{\top}$ to Definition 3.1. With first-layer bias, we have

$$
\boldsymbol{W}_2(\boldsymbol{W}_1\boldsymbol{x} + \boldsymbol{b}_1) = \boldsymbol{W}_2 \begin{pmatrix} \boldsymbol{W}_1 & \boldsymbol{b}_1 \end{pmatrix} \begin{pmatrix} \boldsymbol{x} \\ 1 \end{pmatrix},
\tag{18}
$$

corresponding to $\boldsymbol{G}, \boldsymbol{H} \leftarrow \boldsymbol{W}_2, \begin{pmatrix} \boldsymbol{W}_1 & \boldsymbol{b}_1 \end{pmatrix}^{\top}$.

**Example A.3** (Low-rank adapters (LoRA, Hu et al. (2022))). Fine-tuning tasks with large language models add a trainable low-rank perturbation $\boldsymbol{L} \in \mathbb{R}^{d_1 \times h}, \boldsymbol{R} \in \mathbb{R}^{d_2 \times h}$ to the pre-trained weight $\boldsymbol{W} \in \mathbb{R}^{d_1 \times d_2}$,

$$
\mathrm{LoRA}(\boldsymbol{W}) = \boldsymbol{W} + \boldsymbol{L}\boldsymbol{R}^{\top},
\tag{19}
$$

introducing a $\mathrm{GL}(h)$ symmetry where $\boldsymbol{G}, \boldsymbol{H} \leftarrow \boldsymbol{L}, \boldsymbol{R}$. Yen et al. (2024) propose an invariant way to train the parameters $\boldsymbol{L}, \boldsymbol{R}$ and show that doing so improves the result obtained via LoRA.

# B CONCEPTS AND REVIEW FOR RIEMANNIAN GEOMETRY

Recall that $\overline{\mathcal{M}}$ is the total space: the manifold of parameters of our network. Also, on the total space we have a loss function $\ell : \overline{\mathcal{M}} \to \mathbb{R}$. Useful resources are Lee (2003), Absil et al. (2008), and Boumal (2023).

## B.1 ORBIT OF $x$

A symmetry relation naturally defines an equivalence relation: two points $x, y \in \overline{\mathcal{M}}$ are equivalent under the symmetry, if they can be mapped onto each other by the action,

$$x \sim y \quad \Leftrightarrow \quad \exists g \in \mathcal{G} : y = \psi(g, x). \tag{20}$$

In other words, if we let $\mathrm{orbit}(x) := \{\psi(g, x) \mid g \in \mathcal{G}\}$ be all points on the total space that are reachable from $x$ through the action of $\mathcal{G}$, all points in an orbit are equivalent. Instead of $\mathrm{orbit}(x)$, we will write

$$[x] := \{y \in \overline{\mathcal{M}} \mid y \sim x\} \tag{21}$$

for the symmetry-induced equivalence class $[x]$ of $x \in \overline{\mathcal{M}}$.

Let's further assume that $\bar{\ell}$ is symmetric under $\mathcal{G}$, i.e. for any $x \in \overline{\mathcal{M}}$ and all $g \in \mathcal{G}$, $\bar{\ell}(x) = \bar{\ell}(\psi(g, x))$.

## B.2 QUOTIENT $\mathcal{M}$ AND NATURAL PROJECTION

If we take the quotient of the original parameter space manifold $\overline{\mathcal{M}}$, by the equivalence relation, $\sim$, induced by the symmetries of our neural architecture, we get a quotient $\mathcal{M} = \overline{\mathcal{M}}/\sim$. Under certain conditions, $\mathcal{M}$ is a quotient manifold. The mapping between a point in total space to its equivalence class is called the natural projection:

**Definition B.1.** Let $\pi : \overline{\mathcal{M}} \to \overline{\mathcal{M}}/\sim$, be defined by $\overline{x} \mapsto x$. $\pi$ is called the natural, or canonical projection. We use $\pi(\overline{x})$ to denote $x$ viewed as a point of $\mathcal{M} := \overline{\mathcal{M}}/\sim$.

## B.3 TANGENT SPACE, VERTICAL AND HORIZONTAL SPACES

Tangent vectors on the total space $\overline{\mathcal{M}}$, embedded in a vector space $\mathcal{E}$ can be viewed as tangent vectors to $\mathcal{E}$, but the tangent space of the quotient manifold, $\mathcal{M}$ is not as straightforward. First, note that any element $\bar{\xi} \in T_{\overline{x}}\mathcal{M}$ that satisfies $D\pi(\overline{x})[\bar{\xi}] = \xi$ (where $D$ is the differential) is a candidate for a representation of $\xi \in T_x\mathcal{M}$. These aren't unique, and as we wish to work without any numerical ambiguity we introduce the notions of the vertical and horizontal spaces:

**Definition B.2.** For a quotient manifold $\mathcal{M} = \mathcal{M}/\sim$, the vertical space at $\overline{x} \in \mathcal{M}$ is the subspace $V_{\overline{x}} = T_{\overline{x}}\mathcal{F} = \ker D\pi(x)$ where $\mathcal{F} = \{\overline{y} \in \mathcal{M} : \overline{y} \sim \overline{x}\}$ is the fiber of $\overline{x}$. The complement of $V_{\overline{x}}$ is the horizontal space at $\overline{x}$: $T_{\overline{x}}\overline{\mathcal{M}} = V_{\overline{x}} \oplus H_{\overline{x}}$.

**Definition B.3.** There is only one element $\bar{\xi}_{\overline{x}}$ that belongs to $H_{\overline{x}}$ and satisfies $D\pi(\overline{x})[\bar{\xi}_{\overline{x}}] = \xi$. This unique vector is called the *horizontal lift* of of $\xi$ at $\overline{x}$. We denote the operator that affects the procedure by $\mathrm{lift}_{\overline{x}}(\cdot)$ When the ambient space, $\mathcal{E}$ is a subset of $\mathbb{R}^{n \times p}$, the horizontal space can also be seen as such a subset, providing a convenient matrix representation of *a priori* abstract tangent vectors of $\mathcal{M}$.

## B.4 RIEMANNIAN MANIFOLD

We give our total space $\overline{\mathcal{M}}$ a smooth inner product over its tangent vectors to give a Riemannian manifold.

**Definition B.4.** A Riemannian manifold is a pair $(\mathcal{M}, g)$, where $\mathcal{M}$ is a smooth manifold and $g$ is a Riemannian metric, defined as the inner product on the tangent space $T_x\mathcal{M}$ for each point $x \in \mathcal{M}$, $g_x(\cdot, \cdot) : T_x\mathcal{M} \times T_x\mathcal{M} \to \mathbb{R}$. We also use the notation $\langle \cdot, \cdot \rangle_x$ to denote the inner product.

Note that this definition is not as arcane as it may appear since any smooth manifold admits a Riemannian metric, and we can consider the space of parameters of most neural architectures as constituting a smooth manifold, admitting at least a simple, Euclidean, metric.

The horizontal space can now be defined as the *orthogonal* complement of the vertical space: $H_{\bar{x}} = (V_{\bar{x}})^{\perp} = \{u \in T_{\bar{x}}\overline{\mathcal{M}} : \langle u, v \rangle_x = 0 \text{ for all } v \in V_{\bar{x}}\}$. Additionally, letting $\bar{g}_{\bar{x}}$ denote the metric on $\overline{\mathcal{M}}$, if for every $x \in \mathcal{M}$ and every $\xi_x, \zeta_x$ in $T_x\mathcal{M}$, $\bar{g}_{\bar{x}}(\bar{\xi}_{\bar{x}}, \bar{\zeta}_{\bar{x}})$ does not depend on $\bar{x} \in \pi^{-1}(x)$ then, $g_x(\xi_x, \zeta_x) = \bar{g}_{\bar{x}}(\bar{\xi}_{\bar{x}}, \bar{\zeta}_{\bar{x}})$ defines a valid metric on the quotient manifold $\mathcal{M}$.

## B.5 RIEMANNIAN GRADIENT

**Definition B.5.** If $\bar{f}$ is a smooth scalar field on a Riemannian manifold $\overline{\mathcal{M}}$, then the *gradient* of $\bar{f}$ at $\bar{x}$, $\text{grad}\bar{f}(\bar{x})$ is the unique element of $T_{\bar{x}}\overline{\mathcal{M}}$ such that

$$\langle \text{grad}\bar{f}(\bar{x}), \bar{\xi} \rangle_{\bar{x}} = D\bar{f}(\bar{x})[\bar{\xi}], \forall \bar{\xi} \in T_{\bar{x}}\overline{\mathcal{M}}$$

If $\bar{f}$ is a function on $\overline{\mathcal{M}}$, that induces a function $f$ on a quotient manifold $\mathcal{M}$ of $\overline{\mathcal{M}}$, then we can express the horizontal lift of grad $f$ at $\bar{x}$ as

$$\text{lift}_{\bar{x}}(\text{grad f}) = \text{grad}\bar{f}(\bar{x}).$$

## B.6 GEODESIC CURVES

**Definition B.6.**

(a) Geodesic curves, $\bar{\gamma}$, are the curves of minimal distance between two points on a manifold $\overline{\mathcal{M}}$. The distance along the geodesic is called the *geodesic distance*. If $\mathcal{M}$ is a Riemannian quotient manifold of $\overline{\mathcal{M}}$, with canonical projection $\pi$, and $\bar{\gamma}$ is a geodesic on $\overline{\mathcal{M}}$, then $\gamma = \pi \circ \bar{\gamma}$ is a geodesic curve on $\mathcal{M}$.

(b) Alternatively, geodesics, $\bar{\gamma}(t) = 0$ can be defined as curves from a given point $p \in \overline{\mathcal{M}}$, (i.e., $\bar{\gamma}(0) = p$), with initial *velocity*, $\dot{\bar{\gamma}}(0) = \bar{\xi} \in T_{\bar{p}}\overline{\mathcal{M}}$, such that their *acceleration* is zero (a generalization of Euclidean straight lines). This characterization provides us with the following equation in local coordinates for the geodesic:

$$\frac{d^2\gamma^{\lambda}}{dt^2} + \Gamma^{\lambda}_{\mu\nu}\frac{d\gamma^{\mu}}{dt}\frac{d\gamma^{\nu}}{dt} = 0$$

where $\Gamma^{\lambda}_{\mu\nu}$ are the Christoffel symbols, $\Gamma^{\lambda}_{\mu\nu} = \frac{1}{2}g^{\lambda\sigma}\left(\frac{\partial g_{\sigma\mu}}{\partial x^{\nu}} + \frac{\partial g_{\sigma\nu}}{\partial x^{\mu}} - \frac{\partial g_{\mu\nu}}{\partial x^{\sigma}}\right)$. Additionally, the geodesics can also be derived as the curves that are minima of the energy functional

$$S(\gamma) = \int_a^b g_{\gamma(t)}(\dot{\gamma(t)}, \dot{\gamma(t)})dt$$

This second perspective will prove useful for the geodesics of the attention layers.

If the initial velocity tangent vector, $\xi$, is horizontal then, $\forall t, \dot{\bar{\gamma}}(t) \in H_{\bar{\gamma}(t)}$, that is, if the velocity vector starts out as horizontal, then it will stay horizontal. We call these geodesics, *horizontal geodesics*. The curve $\gamma = \pi \circ \bar{\gamma}$ is a geodesic of the quotient manifold $\mathcal{M}$, with the same length as $\bar{\gamma}$. This also holds the other way, i.e., a geodesic in the quotient manifold can be lifted to a horizontal geodesic in the total space.

# C GEODESIC SHARPNESS: PRACTICAL CONCERNS

## C.1 TRANSFORMERS

Transformers, introduced by Vaswani et al. (2017), consists of multi-headed self-attention and feedforward layers, both wrapped in residual connections and layer normalization. Visual transformers, in addition tend to have convolutional layers.

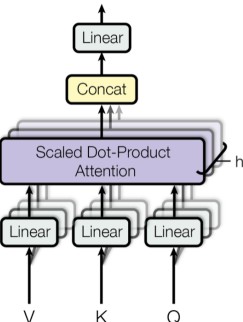

Figure 6: Diagram of multi-headed attention, taken from Vaswani et al. (2017)

Mathematically, focusing for the moment on the multi-headed attention blocks,

$$\text{MultiHead}(Q, K, V) = \big[\text{head}_1, \ldots, \text{head}_h\big] W^o$$

$$\text{where} \quad \text{head}_i = \text{Attention}\big(Q W_i^Q, K W_i^K, V W_i^V\big)$$

where $\text{Attention}(Q, K, V) = \text{softmax}\left(\frac{QK^T}{\sqrt{d_k}}\right) V$. From this we can ascertain the following symmetries:

1) $(W_i^Q, W_i^K) \rightarrow (W_i^Q G^{-1}, W_i^K G^T), \forall G \in \text{GL}_n(d_{\text{head}})$

2) $(W_i^V, W_i^o) \rightarrow (W_i^V G^{-1}, W_i^o G^T), \forall G \in \text{GL}_n(d_{\text{head}})$

where $W_i^o$ are the columns of $W^o$ that are relevant for the matrix multiplication with each $W_i^V$, taking into consideration the head concatenation procedure.

In the full transformer model when solving for geodesic sharpness, for each layer, we apply Eq. 7 to each $(W_i^Q, W_i^K)$ and $(W_i^V, W_i^o)$, using Eq. 16. This results in horizontal vectors $(\bar{\xi}_i^Q, \bar{\xi}_i^K)$ and $(\bar{\xi}_i^V, \bar{\xi}_i^o)$. For the non-attention parameters, $\boldsymbol{w}$, (belonging to fully connected layers, convolutional layers and layer norm), we keep to the recipe of adaptive sharpness, so that $||\bar{\xi}_{\boldsymbol{w}}|| = || \left(\bar{\xi}_{\boldsymbol{w}} \odot |\boldsymbol{w}|^{-1}\right) ||_2$. The norm of the full update vector $\bar{\xi} = \text{concat}(\bar{\xi}_i^Q, \bar{\xi}_i^K, \bar{\xi}_i^V, \bar{\xi}_i^o, \bar{\xi}_{\boldsymbol{w}})$, where a sum over all parameters of the network is implicit, $||\bar{\xi}||^2 = \sum \left(||(\bar{\xi}_i^Q, \bar{\xi}_i^K)||^2 + ||(\bar{\xi}_i^V, \bar{\xi}_i^o)||^2 + ||\bar{\xi}_{\boldsymbol{w}}||^2\right)$.

## C.2 ALGORITHM

Following the lead of Andriushchenko et al. (2023), we use Auto-PDG, proposed in Croce & Hein (2020), but now optimizing the horizontal vector $\bar{\xi}$ instead of the input. In Algorithm 1, $\ell$ is the loss over the batch we are optimizing over, S is the feasible set of horizontal vectors, $\bar{\xi}$, with norm smaller than $\rho$, and $P_S$ is the projection onto this set. $\Gamma$ are the Christoffel symbols for the parameters. $\eta$ and $W$ are fixed hyperparameters, which we keep as in Andriushchenko et al. (2023), and the two conditions in Line 13 can be found in Croce & Hein (2020). The only differences to the algorithm employed to calculate adaptive sharpness are in lines 3, 8, 10, and 12.

## C.3 COMPLEXITY

Geodesic sharpness is slightly more expensive than adaptive sharpness in the following sense: Our approach consists of three steps: 1) perturbing the weights according to Equation 8, 2) optimizing the perturbations with gradient descent, and 3) projecting them onto the feasible set, i.e. horizontal vectors within the geodesic ball with a small enough nerm.

Steps 1) and 2) are also present in adaptive sharpness. Step 1) in our approach is slightly more expensive because we need to evaluate the quadratic form that involves the Christoffel symbols (given by 16); this step introduces $n_{\text{params}}$ weight matrix multiplications, but these are quite efficient. Making the gradients Riemannian, costs another $n_{\text{params}}$ weight matrix multiplications. Neither of these bottleneck our approach. Step 3) requires solving a Sylvester equation to project the direction

---

**Algorithm 1** Auto-PGD

---

1: **Input:** objective function $\ell$, perturbation set $S$, $\bar{\xi}^{(0)}$, initial weights $\boldsymbol{w}^{(0)}$, $\eta$, $N_{\text{iter}}$, $W = \{w_0, \dots, w_n\}$
2: **Output:** $\bar{\xi}_{\max}$, $\ell_{\max}$
3: $\boldsymbol{v}^{(1)} \leftarrow \boldsymbol{w}^{(0)} + \bar{\xi}^{(0)} - \frac{1}{2}\Gamma\bar{\xi}^{(0)}\bar{\xi}^{(0)}$          ▷ Perturb weights according to Eq. 7
4: $\bar{\xi}^{(1)} \leftarrow P_S\left(\bar{\xi}^{(0)} + \eta\nabla_{\bar{\xi}}\ell(\boldsymbol{v}^{(1)})\right)$
5: $\ell_{\max} \leftarrow \max\{\ell(\boldsymbol{w}^{(0)}), \ell(\boldsymbol{v}^{(1)})\}$
6: $\bar{\xi}_{\max} \leftarrow \bar{\xi}^{(0)}$ **if** $\ell_{\max} \equiv \ell(\boldsymbol{w}^{(0)})$ **else** $\bar{\xi}_{\max} \leftarrow \bar{\xi}^{(1)}$
7: **for** $k = 1$ **to** $N_{\text{iter}} - 1$ **do**
8:     $\boldsymbol{v}^{(k+1)} \leftarrow \boldsymbol{w}^{(0)} + \bar{\xi}^{(k)} - \frac{1}{2}\Gamma\bar{\xi}^{(k)}\bar{\xi}^{(k)}$       ▷ Perturb weights according to Eq. 7
9:     **if** $\boldsymbol{w}^{(0)}$ is an attention weight **then**
10:        $g \leftarrow \nabla_{\bar{\xi}}\ell(\boldsymbol{v}^{(k+1)})\boldsymbol{w}^{(0),T}\boldsymbol{w}^{(0)}$      ▷ Make attention gradients Riemannian
11:     **else**
12:        $g \leftarrow \nabla_{\bar{\xi}}\ell(\boldsymbol{v}^{(k+1)}) \odot (\boldsymbol{w}^{(0)})^2$       ▷ Make the other gradients Riemannian
13:     **end if**
14:     $\boldsymbol{z}^{(k+1)} \leftarrow P_S\left(\bar{\xi}^{(k)} + \eta g\right)$
15:     $\bar{\xi}^{(k+1)} \leftarrow P_S\left(\bar{\xi}^{(k)} + \alpha(\boldsymbol{z}^{(k+1)} - \bar{\xi}^{(k)}) + (1-\alpha)(\bar{\xi}^{(k)} - \bar{\xi}^{(k-1)})\right)$
16:     **if** $\ell(\boldsymbol{v}^{(k+1)}) > \ell_{\max}$ **then**
17:        $\bar{\xi}_{\max} \leftarrow \bar{\xi}^{(k+1)}$ and $\ell_{\max} \leftarrow \ell(\boldsymbol{v}^{(k+1)})$
18:     **end if**
19:     **if** $k \in W$ **then**
20:        **if** Condition 1 **or** Condition 2 **then**
21:           $\eta \leftarrow \eta/2$ and $\boldsymbol{w}^{(k+1)} \leftarrow \boldsymbol{w}_{\max}$
22:        **end if**
23:     **end if**
24: **end for**

---

of the updated geodesic back onto the horizontal space. This solve is cubic in $h$ (Kirrinnis, 2001), but $h$ is usually small (e.g. $h = 64$ in the ImageNet and BERT experiments).

On practical transformers, we expect the bottleneck to be the forward and backward propagations, just like in adaptive sharpness.

## D GEODESIC SHARPNESS: SCALAR TOY MODEL

To make our approach explicit, we illustrate it on a NN with two scalar parameters $G$ and $H$, square loss, and a single (scalar) training point $(x, y)$. For this example, everything is analytically tractable. We also contrast our sharpness measure with previously proposed ones to highlight its invariance.

Since we require full column-rank, our parameter space is $\mathcal{M} = \mathbb{R}_* \times \mathbb{R}_*$ with $\mathbb{R}_* = \mathbb{R} \setminus \{0\}$.

**Metric** At a point $(G, H) \in \mathcal{M}$, for two tangent vectors $\eta = (\eta_G, \eta_H)$, $\nu = (\nu_G, \nu_H) \in T_{(G,H)}\mathcal{M}$, we have

$$g\left[(\eta_G, \eta_H), (\nu_G, \nu_H)\right] = \frac{\eta_G\nu_G}{G^2} + \frac{\eta_H\nu_H}{H^2} = \eta^\top \underbrace{\begin{pmatrix} \frac{1}{G^2} & 0 \\ 0 & \frac{1}{H^2} \end{pmatrix}}_{g_{kl}} \nu \tag{22}$$

We denote the inverse metric by $g^{kl} = \begin{pmatrix} G^2 & 0 \\ 0 & H^2 \end{pmatrix}$

**Horizontal space** $H_{(G,H)} = \{(\eta_G, \eta_H) \in T_{(G,H)}\mathcal{M} \mid \frac{\eta_G}{G} = \frac{\eta_H}{H}\}$

**Geodesics** To compute the geodesics on the quotient space, we need the Christoffel symbols $\Gamma^i_{km}$.

Using a coordinate system $(p^1, p^2) = (G, H)$, we have the following equation for a geodesic $\gamma(t) = (\gamma_G(t), \gamma_H(t))$, with initial conditions $\gamma(0) = (G_0, H_0)$ and $\dot{\gamma}(0) = (\eta_{G_0}, \eta_{H_0})$

$$\frac{d^2\gamma_G}{dt^2} + \Gamma^1_{11}\left(\frac{d\gamma_G}{dt}\right)^2 = 0$$

and similarly for $H$ with $\Gamma^2_{22}$ instead of $\Gamma^1_{11}$.

The Christoffel symbols can be found using the metric, $g$, and its inverse. Using the Einstein notation and denoting the inverse of $g$ by the use of upper indices:

$$\Gamma^i_{kl} = \frac{1}{2}g^{im}\left(\frac{\partial g_{mk}}{\partial x^l} + \frac{\partial g_{ml}}{\partial x^k} - \frac{\partial g_{kl}}{\partial x^m}\right)$$

Then

$$\Gamma^1_{11} = \frac{1}{2}g^{1m}\left(\frac{\partial g_{m1}}{\partial p^1} + \frac{\partial g_{m1}}{\partial p^1} - \frac{\partial g_{kl}}{\partial p^m}\right) = -\frac{1}{G}$$

$$\Gamma^2_{22} = -\frac{1}{H}$$

All other Christoffel symbols are 0. Our geodesic equations then become (we omit the derivation for H, which is identical but with $G \leftrightarrow H$)

$$\frac{d^2\gamma_G}{dt^2} - \frac{1}{\gamma_G}\left(\frac{d\gamma_G}{dt}\right)^2 = 0$$

This ODE has the (unique) solution $\gamma_G(t) = A_G\exp(B_G t)$. Taking into account the initial conditions, $A_G = G_0, A_H = H_0$ and due to the definition of the horizontal space, $B_G = \frac{\eta_G}{G_0}$ and $B_H = \frac{\eta_H}{H_0}$, this becomes

$$\gamma(t) = \left(G_0\exp\left(\frac{\eta_G}{G_0}t\right), H_0\exp\left(\frac{\eta_H}{H_0}t\right)\right)$$

One important detail to note is that these geodesics are not complete, that is, not all two points can be connected by a geodesic. Points with different signs can not be connected, which makes sense since we excluded the origin from the acceptable parameters and in 1D we need to cross it to connect points with differing signs. All points that lie in the same quadrant as $(G_0, H_0)$ can be connected through a geodesic.

Putting it all together

$$S^\rho_{\max}((G_0, H_0)) = \left[\max_{||B|| \leq \rho} x^2 G_0^2 H_0^2(\exp(4B) - 1) - 2yxG_0H_0(\exp(2B) - 1)\right], \quad (23)$$

Letting $y_0 = G_0H_0x$, this becomes:

$$S^\rho_{\max}((G_0, H_0)) = \left[\max_{||B|| \leq \rho} y_0^2(\exp(4B) - 1) - 2yy_0(\exp(2B) - 1)\right], \quad (24)$$

Since $\eta_H$ is completely determined by $\eta_G$ we can ignore the maximization over it.

Since in practice we'll take $\rho \ll 1$, we Taylor expand to get

$$S^\rho_{\max} \approx 4\rho|y_0||y - y_0|$$

This presents an issue when the residual, $|y - y_0|$, is zero, so we also can expand to second order, to get, when $|y - y_0| \approx 0$

$$S^\rho_{\max} \propto \rho^2|y_0||y - 2y_0| = 2\rho^2 y_0^2$$

This is, up to constants, just $||G \odot H||_2^2$. This is also invariant to $GL_1$ transformations, as expected.

Very close to the minimum we only capture (second-order in $\rho$) properties of the network, a bit further away from it we capture a (first-order in $\rho$) mix of data and network properties.

**Comparison with more traditional measures**   The local average and worst case Euclidean sharpness (at a minimum) are

$$S_{\text{avg}} = \text{Tr}\,\nabla^2 L_S = G^2 + H^2$$
$$S_{\text{max}} = \lambda_{\text{max}}(\nabla^2 L_S) = G^2 + H^2$$

Adaptive sharpness is defined as

$$S_{\text{avg}}^\rho(w, c) = \mathbb{E}_{S \sim \mathbb{P}_m}\left[L_S(w + \delta) - L_S(w)\right], \quad \delta \sim \mathcal{N}(0, \rho^2 \text{diag}(c^2))$$
$$S_{\text{max}}^\rho(w, c) = \mathbb{E}_{S \sim \mathbb{P}_m}\left[\max_{\|\delta \odot c^{-1}\|_p \leq \rho} L_S(w + \delta) - L_S(w)\right],$$

By picking $c$ very carefully one can get

$$S_{\text{avg}}^\rho(w, c) = |GH|$$
$$S_{\text{max}}^\rho(w, c) = |GH|$$

By contrast, in our approach there is no need for careful hyperparameter choices

**Geodesic flatness with more data points**   How does the geodesic flatness look like with more data points?

$$L_S(G, H) = \frac{1}{n}\sum_{i=1}^{n}(GHx_i - y_i)^2$$

which leads to (defining $y_i^0 = GHx_i$):

$$S_{\text{max}}^\rho = \max_B \frac{1}{n}\sum_{i=1}^{n}\left[(y_i^0)^2\left(\exp\left(\frac{B}{|B|}2\sqrt{2}\rho\right) - 1\right) - 2yy_i^0\left(\exp\left(\frac{B}{|B|}\sqrt{2}\rho\right) - 1\right)\right] \quad (25)$$

Taylor expanding (in $\rho$) once more, we see that

$$S_{\text{max}}^\rho \approx \max_B \frac{1}{n}\sum_{i=1}^{n}\left[2\sqrt{2}\rho\frac{B}{|B|}y_i^0(y_i^0 - y) + 2\rho^2(y_i^0)^2\right] \quad (26)$$

Which $B$ maximizes Eq. 26, depends on the sign of $\sum_{i=1}^{n}\left[y_i^0(y_i^0 - y)\right]$: $B < 0$ if the sum is negative, the reverse if the opposite is true.

## D.1   TRADITIONAL FLATNESS

In Figure 7 we extend Figure 1 to include the trace of the Hessian, both Euclidean and Riemannian. The reason we picked the trace of the network Hessian is that it is a quantity that can be used to quantify flatness. We plot, for the scalar toy model, the level sets of: a) the loss function; b) the Euclidean and Riemannian gradient; c) the traces of the Euclidean and Riemannian network Hessian. Of note is that a) the Riemannian version of the gradient and Hessian have the same level set geometry as the loss function; b) both the Riemannian gradient norm and the trace of the Riemannian hessian have smaller values throughout than their Euclidean equivalents; c) the trace of Riemannian hessian actually reaches 0 when at the local minimum, whereas the Euclidean hessian actually attains its highest value there; d) the Euclidean trace of the hessian is unable to distinguish between a minimum and a maximum whereas the Riemannian trace can actually do so. Even for simple flatness measures, correcting for the quotient geometry can provide a much clearer picture.

## E   GEODESIC SHARPNESS: DIAGONAL NETWORKS IN FULL GENERALITY

In this appendix, we continue the analysis from the main body and extend it to

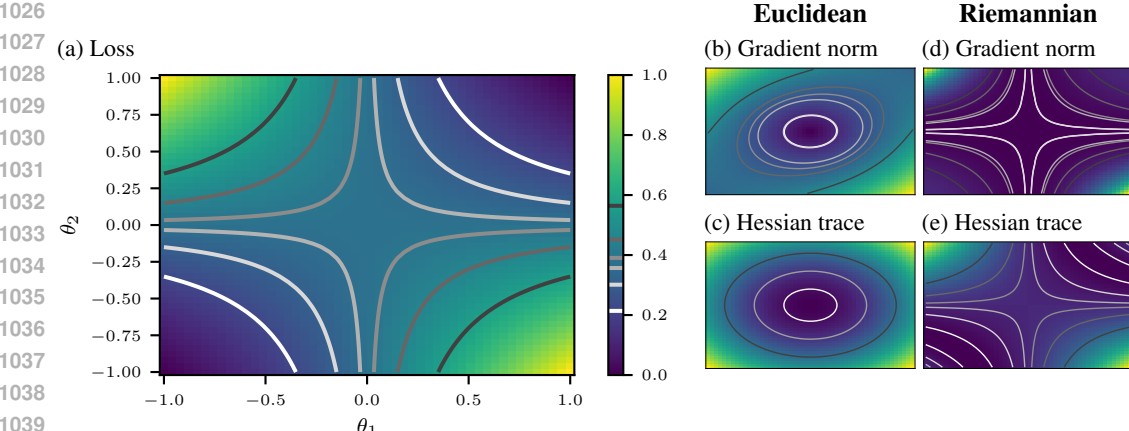

Figure 7: **Quantities from the Riemannian quotient manifold respect the loss landscape's symmetry; Euclidean quantities do not.** We use a synthetic least squares regression task with a two-layer NN $x \mapsto \theta_2 \theta_1 x$ with scalar parameters $\theta_i \in \mathbb{R}$ and input $x \in \mathbb{R}$. The NN is re-scale invariant, i.e. has GL(1) symmetry: For any $\alpha \in \mathbb{R} \setminus \{0\}$, the parameters $(\theta_1', \theta_2') = (\alpha^{-1}\theta_1, \alpha\theta_2)$ represent the same function. (a) The loss function inherits this symmetry and has hyperbolic level sets. (b,c) The Euclidean gradient norm does not share the loss function's geometry and changes throughout an orbit where the NN function remains constant. (d,e) The Riemannian gradient norm follows the loss function's symmetry and remains constant throughout an orbit, i.e. it does not suffer from ambiguities for two points in parameter space that represent the same NN function. All quantities were normalized to $[0; 1]$ and we fixed six points in parameter space and computed the level sets running through them to illustrate the geometry.

CASE C): $\boldsymbol{r} \neq 0$ AND WE NEED BOTH FIRST AND SECOND ORDER TERMS    In this case, Eq.12 needs to be considered in full, and we solve the maximization problem using Lagrange multipliers. The Lagrangian will be

$$\mathcal{L} = -4\boldsymbol{B}^T\boldsymbol{r} - 4\boldsymbol{B}^T\boldsymbol{D}_{\boldsymbol{\beta}_0, \boldsymbol{\beta}_*}\boldsymbol{B} + \lambda(\boldsymbol{B}^T\boldsymbol{B} - \rho^2)$$

The KKT conditions then are

$$\frac{\partial \mathcal{L}}{\partial \boldsymbol{B}} = -4\boldsymbol{r} - 8\boldsymbol{D}_{\boldsymbol{\beta}_0, \boldsymbol{\beta}_*}\boldsymbol{B} + 2\lambda\boldsymbol{B} = 0 \tag{27}$$

$$\lambda(\boldsymbol{B}^T\boldsymbol{B} - \rho^2) = 0 \tag{28}$$

$$\lambda \geq 0 \tag{29}$$

If the constraint is not active, then $\lambda = 0$ and

$$\boldsymbol{B}_* = -\frac{1}{2}\boldsymbol{D}_{\boldsymbol{\beta}_0, \boldsymbol{\beta}_*}^{-1}\boldsymbol{r}$$

In practice, unless $\rho$ is large the constraint will always be active, in which case 27 becomes

$$-4\boldsymbol{r} - 8\boldsymbol{D}_{\boldsymbol{\beta}_0, \boldsymbol{\beta}_*}\boldsymbol{B} + 2\lambda(\boldsymbol{B}) = 0$$
$$(\boldsymbol{B}^T\boldsymbol{B} - \rho^2) = 0$$
$$\lambda \geq 0$$

this then becomes

$$\boldsymbol{B}_* = 2\left(\lambda I - 4\boldsymbol{D}_{\boldsymbol{\beta}_0, \boldsymbol{\beta}_*}\right)^{-1} \boldsymbol{r}$$

$$4\sum_i^d \frac{(\boldsymbol{r}^i)^2}{\left(\lambda - 4(\boldsymbol{\beta}_0^i(\boldsymbol{\beta}_0^i + \boldsymbol{r}'))\right)^2} = \rho^2$$

$$\lambda \geq 0$$

## F  GEODESIC SHARPNESS: HERE BE METRICS

### F.1  $GL_1$ SYMMETRY AND ADAPTIVE SHARPNESS

What happens if instead of a general $GL_n$ symmetry, we factor out a $GL_1$ re-scaling symmetry? That is, we identify, element-wise, $(\bar{x}, \bar{y}) \sim (\bar{x}'\bar{y}')$ if $\exists \alpha \in \mathbb{R}_* = \mathbb{R} \setminus \{0\}$ s.t. $\bar{x} = \alpha\bar{x}'$ and $\bar{y} = \alpha^{-1}\bar{y}$.

This is the symmetry present in diagonal networks, and so we utilize the metric given by Eq. 9, reproduced below for convenience of the reader:

$$g\left[(\eta_{\boldsymbol{u}}, \eta_{\boldsymbol{v}}), (\nu_{\boldsymbol{u}}, \nu_{\boldsymbol{v}})\right] = \sum_{i=1}^d \frac{\eta_{\boldsymbol{u}}^i \nu_{\boldsymbol{u}}^i}{(\boldsymbol{u}^i)^2} + \frac{\eta_{\boldsymbol{v}}^i \nu_{\boldsymbol{v}}^i}{(\boldsymbol{v}^i)^2} \tag{30}$$

Note that this metric is equivalent to the following metric:

$$g\left[(\eta_{\boldsymbol{u}}, \eta_{\boldsymbol{v}}), (\nu_{\boldsymbol{u}}, \nu_{\boldsymbol{v}})\right] = g\left[(\eta_{\boldsymbol{u}}/|\boldsymbol{u}|, \eta_{\boldsymbol{v}}/|\boldsymbol{v}|), (\nu_{\boldsymbol{u}}/|\boldsymbol{u}|, \nu_{\boldsymbol{v}}/|\boldsymbol{v}|)\right]_{\text{euc}} \tag{31}$$

where $g_{\text{euc}}$ is the usual Euclidean metric and the division is taken to be element-wise. Denoting the concatenation of all tangent vectors by $\xi$, the concatenation of all parameters by $\boldsymbol{w}$, we have $||\xi|| = ||\xi/|\boldsymbol{w}|||_2$.

In this situation Eq. 6 becomes ($\gamma$ denotes our geodesics as usual)

$$S_{\max}^\rho(w, c) = \mathbb{E}_{\mathbb{S} \sim \mathbb{D}} \left[ \max_{||\xi/|\boldsymbol{w}|||_2 \leq \rho} L_S(\bar{\gamma}_{\bar{\xi}}(1)) - L_S(\bar{\gamma}_{\bar{\xi}}(0)) \right], \tag{32}$$

If we then ignore the corrections induced by the geometry of the metric on the geodesics, i.e., take $\bar{\gamma}_{\bar{\xi}}(1) = \bar{\gamma}_{\bar{\xi}}(0) + \bar{\xi} = \boldsymbol{w} + \bar{\xi}$, then we get

$$S_{\max}^\rho(w, c) = \mathbb{E}_{\mathbb{S} \sim \mathbb{D}} \left[ \max_{||\xi/|\boldsymbol{w}|||_2 \leq \rho} L_S(\boldsymbol{w} + \xi) - L_S(\boldsymbol{w}) \right] \tag{33}$$

which is exactly the formula for adaptive sharpness.

### F.2  A DIFFERENT METRIC

The metric given by Eq. 15 is not the only metric that allows for the construction of the quotient manifold. One such family is that of metrics related to metric 15 by re-scaling and constant shifts result: in appendix I.3 we show that the result in the same geodesic sharpness. We are aware of only one other metric that respects the $GL_n$ symmetry and is described in the literature Mishra et al. (2012):

$$\langle \bar{\eta}, \bar{\zeta} \rangle_x = \text{Tr}\left((\boldsymbol{H}^\top \boldsymbol{H})\bar{\eta}_{\boldsymbol{G}}^\top \bar{\zeta}_{\boldsymbol{G}} + (\boldsymbol{G}^\top \boldsymbol{G})\bar{\eta}_{\boldsymbol{H}}^\top \bar{\zeta}_{\boldsymbol{H}}\right), \tag{34}$$

which mixes the components.

**Horizontal space:** $\mathcal{H}_x \overline{\mathcal{M}} = \{(\bar{\xi}_{\boldsymbol{G}}, \bar{\xi}_{\boldsymbol{H}}) \mid \boldsymbol{G}^T \bar{\xi}_{\boldsymbol{G}} \boldsymbol{H}^T \boldsymbol{H} = \boldsymbol{G}^T \boldsymbol{G} \xi_{\boldsymbol{H}}^T \boldsymbol{H}, \bar{\xi}_{\boldsymbol{G}} \in \mathbb{R}^{n \times r}, \bar{\xi}_{\boldsymbol{H}} \in \mathbb{R}^{m \times r}\}$.

**Projection onto the horizontal space:** $\Pi_{\bar{x}}(\bar{\xi}_{\bar{x}}) = \left(\bar{\xi}_{\boldsymbol{G}} + \boldsymbol{G}\Lambda, \bar{\xi}_{\boldsymbol{H}}^T - \boldsymbol{H}\Lambda^T\right)$, with $\Lambda = \frac{1}{2}\left(\bar{\xi}_{\boldsymbol{H}}^T \boldsymbol{H}(\boldsymbol{H}^T \boldsymbol{H})^{-1} - (\boldsymbol{G}^T \boldsymbol{G})^{-1} \boldsymbol{G}^T \bar{\xi}_{\boldsymbol{G}}\right)$

**Geodesics:**   Applying the same procedure as in I.2, we get the following geodesic equations

$$\ddot{G}H^T H + \dot{G}(\dot{H}^T H + H^T \dot{H}) - G\dot{H}^T \dot{H} = 0 \tag{35}$$

$$\ddot{H}G^T G + \dot{H}(\dot{G}^T G + G^T \dot{G}) - H\dot{G}^T \dot{G} = 0 \tag{36}$$

### F.2.1   EMPIRICAL VALIDATION

We employ metric (34) to study the same collection of BERT models that we studied using metric 15 in the main text. The experimental setup is identical to that of the main text.

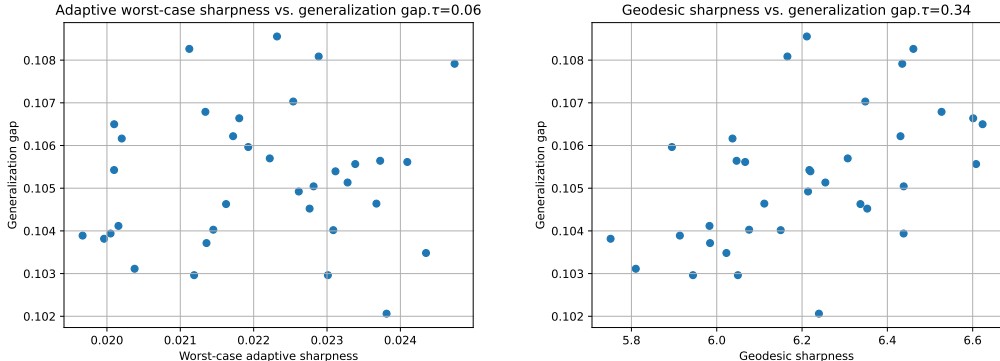

Figure 8: The generalization gap on the MNLI dev matched set (Williams et al., 2018) vs. worst-case adaptive sharpness with metric (34) (left) and geodesic sharpness (right), is shown for 35 models from (McCoy et al., 2020)

Interestingly, we found that this metric actually improved our results on BERT, leading to a $\tau$ of 0.34. We suspect this is due to more favourable numerics: this new metric is of the form $Tr[(H^T H)\bar{\xi}^T \bar{\xi}]$ as opposed to $Tr[(G^T G)^{-1}\bar{\xi}^T \bar{\xi}]$, side-stepping issues with possibly large singular values. Additionally, this metric does not require a Sylvester solver in order to project into the horizontal space, making this metric also faster to iterate with. Mishra et al. (2012) studied this metric in the context of low-rank matrix completion, where they argue that it is better-tuned to the squared losses commonly used in that particular setting; something similar could be at play here as well.

## G   GEODESIC SHARPNESS: ABLATIONS

In this appendix we conduct ablation studies on geodesic sharpness 6. There are two main components to our recipe that differ from adaptive sharpness: a) the norm $||\bar{\xi}||$; b) the weight update formula, which instead of the usual $\boldsymbol{w}^i = \boldsymbol{w}^i + \bar{\xi}$ takes into account the curvature induced by the parameter space symmetries $\boldsymbol{w}^i = \boldsymbol{w}^i + \bar{\xi}^i - \frac{1}{2}\Gamma_{kl}^i \bar{\xi}^k \bar{\xi}^l$. Below we turn off these components one by one and re-compute the resulting sharpness on MNLI using the BERT models described in Section 5.3.2.

**Metric (15)**   In Figure 9 we show the results for our ablation studies using metric (15). The norm component is much more impactful than the second-order weight corrections. Turning off the second-order weight corrections results in a small performance drop only.

**Metric (34)**   In Figure 10 we show the results for our ablation studies using metric (34). The norm component is still much more impactful than the second-order weight corrections for this metric, but now the second-order weight corrections are essential, and without them sharpness loses a considerable amount of predictive power.

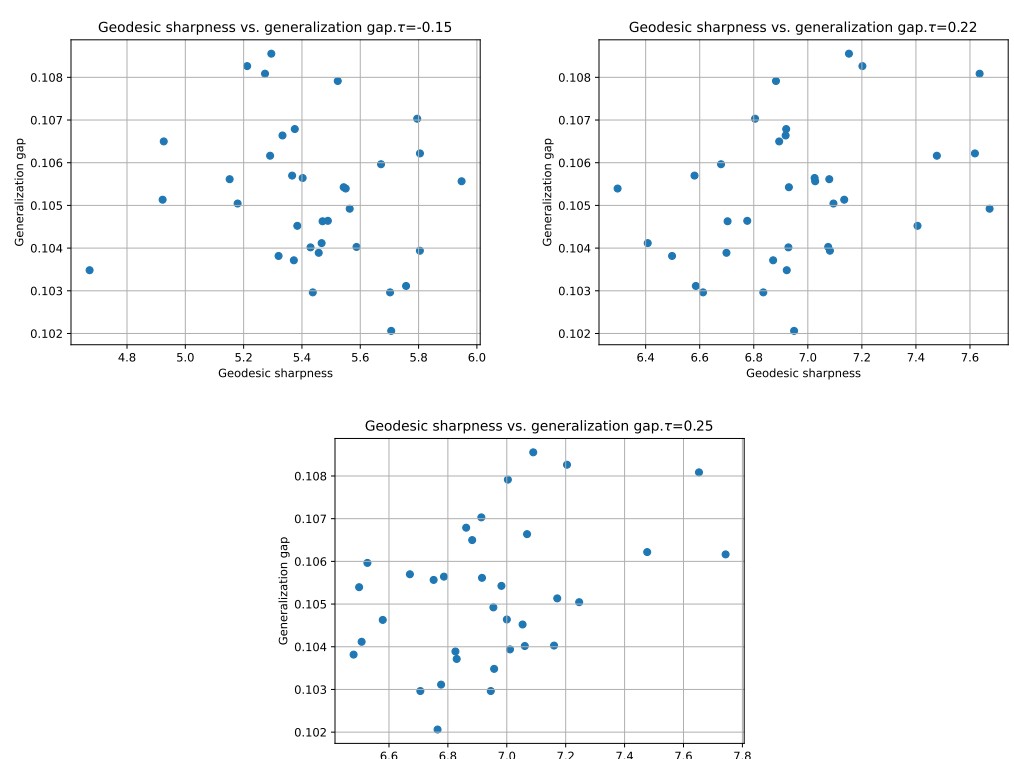

Figure 9: The generalization gap on the MNLI dev matched set (Williams et al., 2018) vs. worst-case adaptive sharpness with metric (15) is shown for 35 models from (McCoy et al., 2020). On the left we plot the results when we turn off the corrected norm, and on the right when we turn off the second-order weight corrections. Bottom are the results with no ablations.

# H    GEODESIC SHARPNESS: RANKS AND RELAXATION

## H.1    RANKS: HOW NATURAL IS ASSUMPTION 5.1?

In general, in non-linear networks there is a tendency towards low-rank representations, which might make Assumption 5.1 seem excessive and counter to realistic situations. However, while the learned $W_Q W_K^T$ tend to be low-rank, $W_Q$ and $W_K$ (on which Assumption 5.1 ought to apply) themselves are usually high/full (column) rank Yu & Wu (2023).

## H.2    RELAXATION

Due to the definition of metric 15, we need to invert matrices of the type of $W_Q^T W_Q$. When these are not full-rank, numerical stability can suffer. Due to floating-point precision rounding errors, in practice $W_Q^T W_Q$ is always invertible, but sometimes the inverted matrices have huge singular values. To combat this, we introduce a relaxation parameter, so that $W_Q^T W_Q \rightarrow W_Q^T W_Q + \epsilon I_h$, which dampens the resulting singular values. Although we cannot take it to be exactly zero, as long as it is small enough, numerical stability is improved and the results remain roughly the same. We study the effects of varying this parameter on our results empirically below (Figure 11), using the same setup as in Section 5.3.2. The results are not significantly affected by the variation of this parameter.

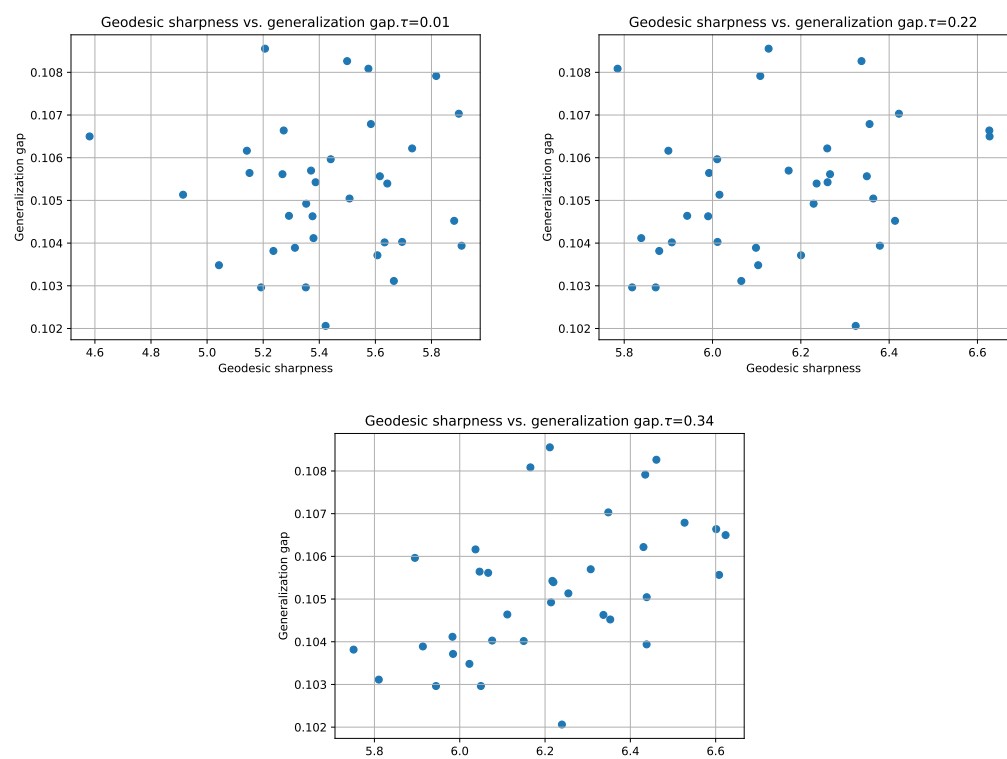

Figure 10: The generalization gap on the MNLI dev matched set (Williams et al., 2018) vs. worst-case adaptive sharpness with metric (34) is shown for 35 models from (McCoy et al., 2020). On the left we plot the results when we turn off the corrected norm, and on the right when we turn off the second-order weight corrections. Bottom are the results with no ablations.

# I ADDITIONAL DERIVATIONS AND PROOFS

## I.1 PROOF THAT EQ. 15 DEFINES A VALID RIEMANNIAN METRIC

Eq. 15 defines a valid metric on the total space $\overline{\mathcal{M}}$ if it is smooth, and for each point $(\bar{G}, \bar{H}) \in \overline{\mathcal{M}}$ it defines a valid inner product on the tangent space $T_{(\bar{G}, \bar{H})}\overline{\mathcal{M}}$. That it is smooth is obvious, so we show that $\langle \bar{\eta}, \bar{\zeta} \rangle_{(\bar{G}, \bar{H})} = \text{Tr}\left( (\boldsymbol{G}^\top \boldsymbol{G})^{-1} \bar{\eta}_{\boldsymbol{G}}^\top \bar{\zeta}_{\boldsymbol{G}} + (\boldsymbol{H}^\top \boldsymbol{H})^{-1} \bar{\eta}_{\boldsymbol{H}}^\top \bar{\zeta}_{\boldsymbol{H}} \right)$ defines a valid inner product:

(i) *Symmetry* $\langle \bar{\eta}, \bar{\zeta} \rangle = \langle \bar{\zeta}, \bar{\eta} \rangle$: omitting the $\boldsymbol{H}$ term as it is identical, $\langle \bar{\eta}, \bar{\zeta} \rangle = \text{Tr}\left( (\boldsymbol{G}^\top \boldsymbol{G})^{-1} \bar{\eta}_{\boldsymbol{G}}^\top \bar{\zeta}_{\boldsymbol{G}} \right) = \text{Tr}\left( (\boldsymbol{G}^\top \boldsymbol{G})^{-1} \bar{\zeta}_{\boldsymbol{G}}^\top \bar{\eta}_{\boldsymbol{G}} \right) = \langle \bar{\zeta}, \bar{\eta} \rangle$ ;

(ii) *Bilinearity* $\langle a\bar{\eta} + b\bar{\zeta}, \bar{\lambda} \rangle = a\langle \bar{\eta}, \bar{\lambda} \rangle + b\langle \bar{\zeta}, \bar{\lambda} \rangle = \langle \bar{\lambda}, a\bar{\eta} + b\bar{\zeta} \rangle$: follows by linearity of the trace;

(iii) *Positive Definiteness* $\langle \bar{\eta}, \bar{\eta} \rangle \geq 0$: using assumption 5.1, $\boldsymbol{G}^T \boldsymbol{G}$ is invertible and is positive-definite; this means that $(\boldsymbol{G}^T \boldsymbol{G})^{-1}$ is also positive-definite, and so $\langle \bar{\eta}, \bar{\eta} \rangle \geq 0$, with equality only when $\bar{\eta} = 0$.

## I.2 DERIVATION OF THE GEODESIC CORRECTIONS FOR ATTENTION

We apply the Euler-Lagrange formalism to the energy functional to derive the geodesic equation on the attention quotient manifold, and hence $\Gamma^i_{kl} \bar{\xi}^k_{\boldsymbol{G}} \bar{\xi}^l_{\boldsymbol{G}}$, remembering that geodesics, in local coordinates, obey the equation $\frac{d^2\gamma^i}{dt^2} + \Gamma^i_{kl} \frac{d\gamma^k}{dt} \frac{d\gamma^l}{dt} = 0$.

Starting from

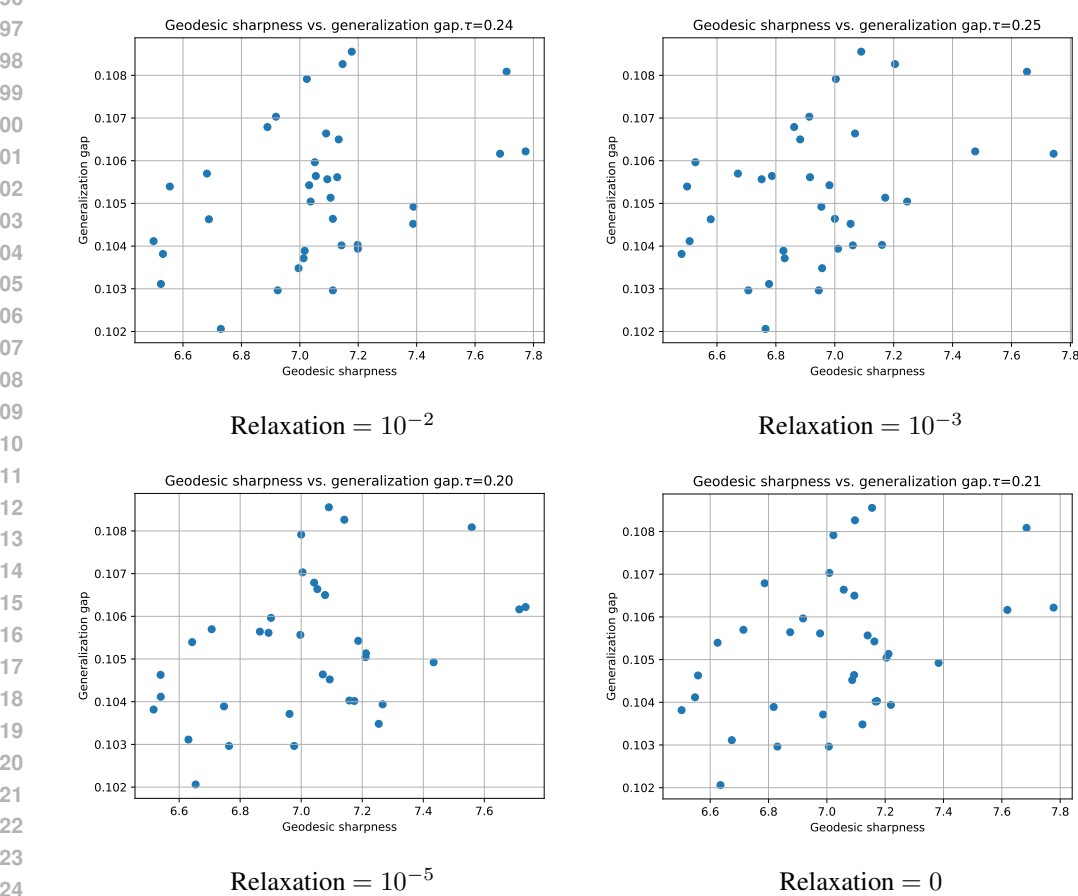

Figure 11: The generalization gap on the MNLI dev matched set (Williams et al., 2018) vs. worst-case adaptive sharpness (left) and geodesic sharpness (right), is shown for 35 models from (McCoy et al., 2020). The difference between each plot is only in the relaxation parameter. The results stay broadly the same.

$$E(\gamma) = \int_0^1 \mathcal{L} \, dt = \int_0^1 \langle \dot{\gamma}(t), \dot{\gamma}(t) \rangle_{\gamma(t)} dt \tag{37}$$

$$= \int_0^1 \left[ \mathrm{Tr}\big(\gamma_{\boldsymbol{G}}(t)^T \gamma_{\boldsymbol{G}}(t)\big) \dot{\gamma}_{\boldsymbol{G}}(t)^T \dot{\gamma}_{\boldsymbol{G}}(t) + \mathrm{Tr}\big(\gamma_{\boldsymbol{H}}(t)^T \gamma_{\boldsymbol{H}}(t)\big) \dot{\gamma}_{\boldsymbol{H}}(t)^T \dot{\gamma}_{\boldsymbol{H}}(t) \right] dt \tag{38}$$

,

The Euler-Lagrange equation, for the $\boldsymbol{G}$ part only, reads

$$\frac{d}{dt}\left(\frac{\partial \mathcal{L}}{\partial \dot{\boldsymbol{G}}}\right) - \frac{\partial \mathcal{L}}{\partial \boldsymbol{G}} = 0 \tag{39}$$

We have

$$\frac{\partial \mathcal{L}}{\partial \boldsymbol{G}} = -2\boldsymbol{G}\left(\boldsymbol{G}^T\boldsymbol{G}\right)^{-1}\left(\dot{\boldsymbol{G}}^T\dot{\boldsymbol{G}}\right)\left(\boldsymbol{G}^T\boldsymbol{G}\right)^{-1} \tag{40}$$

$$\frac{d}{dt}\left(\frac{\partial \mathcal{L}}{\partial \dot{\boldsymbol{G}}}\right) = 2\ddot{\boldsymbol{G}}\left(\boldsymbol{G}^T\boldsymbol{G}\right)^{-1} - 2\dot{\boldsymbol{G}}\left(\boldsymbol{G}^T\boldsymbol{G}\right)^{-1}\left(\dot{\boldsymbol{G}}^T\boldsymbol{G} + \boldsymbol{G}^T\dot{\boldsymbol{G}}\right)\left(\boldsymbol{G}^T\boldsymbol{G}\right)^{-1} \tag{41}$$

So that Eq. 39 becomes:

$$\ddot{\boldsymbol{G}} - \dot{\boldsymbol{G}} \left(\boldsymbol{G}^T\boldsymbol{G}\right)^{-1} \left(\dot{\boldsymbol{G}}^T\boldsymbol{G} + \boldsymbol{G}^T\dot{\boldsymbol{G}}\right) + \boldsymbol{G} \left(\boldsymbol{G}^T\boldsymbol{G}\right)^{-1} \left(\dot{\boldsymbol{G}}^T\dot{\boldsymbol{G}}\right) = 0 \tag{42}$$

From which we read

$$\Gamma^i_{kl}\bar{\xi}^k_{\boldsymbol{G}}\bar{\xi}^l_{\boldsymbol{G}} = \left[-\bar{\xi}\left(\boldsymbol{G}^T\boldsymbol{G}\right)^{-1}\left(\bar{\xi}^T\boldsymbol{G} + \boldsymbol{G}^T\bar{\xi}\right) + \boldsymbol{G}\left(\boldsymbol{G}^T\boldsymbol{G}\right)^{-1}\left(\bar{\xi}^T\bar{\xi}\right)\right]^i \tag{43}$$

## I.3 METRICS RELATED BY SCALING AND CONSTANTS

If $g$ is a metric and $g_{\text{scaled}} = Cg + D$, then from Eq. 37 and Eq. 39 we see that the geodesics induced by $g_{\text{scaled}}$ are the same as those induced by $g$. The geodesic sharpness induced by $g_{\text{scaled}}$ is

$$S^\rho_{\max}(w) = \mathbb{E}_{\mathbb{S}\sim\mathbb{D}}\left[\max_{||\bar{\xi}||_{\bar{\gamma}_{\text{scaled}}}\leq\rho} L_S(\bar{\gamma}_{\bar{\xi};\text{scaled}}(1)) - L_S(\bar{\gamma}_{\bar{\xi};\text{scaled}}(0))\right] =$$

$$= \mathbb{E}_{\mathbb{S}\sim\mathbb{D}}\left[\max_{C||\bar{\xi}||_{\bar{\gamma}}+D\leq\rho} L_S(\bar{\gamma}_{\bar{\xi}}(1)) - L_S(\bar{\gamma}_{\bar{\xi}}(0))\right],$$

$$= \mathbb{E}_{\mathbb{S}\sim\mathbb{D}}\left[\max_{||\bar{\xi}||_{\bar{\gamma}}\leq\rho'} L_S(\bar{\gamma}_{\bar{\xi}}(1)) - L_S(\bar{\gamma}_{\bar{\xi}}(0))\right],$$

So they are the same up to some re-definition of the hyperparameter $\rho$.