# OpenReview forum: "Hide & Seek: Transformer Symmetries Obscure Sharpness & Riemmanian Geometry Finds It"
_ICLR.cc/2025/Conference — Submitted to ICLR 2025_

### Official Review · Reviewer_qb83 · 2024-10-30

**Soundness:** 4
**Presentation:** 2
**Contribution:** 3
**Rating:** 6
**Confidence:** 3

**Summary:**

This paper generalizes the (adaptive) sharpness concept by considering both the first-order perturbation and the (second-order) curvature of the loss landscape, presented under a Riemannian geodesic framework. It aims at solving the inconsistency in the literature on the correlation between sharpness and generalization, and addresses the failure of existing sharpness in measuring the generalization of Transformers with larger symmetry, explained by the scaling/rescaling symmetry of the parameter space. The symmetry is formalized as the quotient group. Experiments on both a toy diagonal network and fine-tuned vision transformers show that the geodesic sharpness has a significant correlation with the generalization gap.

**Strengths:**

The motivation is strong, the theoretical discussion is rich and sound, and the curvature-aware sharpness significantly depends on the generalization gap. I appreciate the insights of the Attention symmetry and the strong theoretical connections.

**Weaknesses:**

- My main concern is the practical effectiveness of the proposed method: does it really improve existing adaptive sharpness? In Figure 4 the improvements are not quite visible, and the parameter $\tau$ is neither explained nor aligned by column.
- The experiments are not described in detail, which affects reproducibility. For self-containedness, the method used in the experiment from Wortsman et al. (2022) could be explained better in detail. For example, what are the hyperparameters, how are the models trained, what is the precise algorithm, etc.?
- It is questionable whether the attention's symmetry argued by the author is indeed the practical reason rather than a hypothesis. It would be helpful if the author could give a stronger argument or some illustrations. To give an example, ensure that the scaling symmetry causes irregularity in pre-trained models by measuring the conditional number, and modulating out this symmetry indeed corrects the sharpness---or any other way you prefer.
- I believe the paper's writing could benefit greatly by being more concise. Many sentences in the introduction seem to wander around and are not precisely to the point.
- Minor issues: Figures 2, 3, and 4 are not referenced.

**Questions:**

- I don't understand the role of the $\tau$ variable in the result figures. For example, could you explain how it is chosen, and how it relates to the results?
- I don't understand why in Figure 3 (left), the average-case adaptive sharpness is claimed to reveal a correlation with the test loss (generalization gap). To me, it looks not obvious, and the dependency is reversed.
- $c$ inconsistently appears in Appendix D. Do the experiments of geodesic sharpness also choose a vector $c$ by taking $c=|w|$ to normalize each parameter tensor? If not how to guarantee fair comparison? In Appendix D $c$ is preset, but I don't understand whether the geodesic is invariant over scaling?
- I wonder if normalizing QK embeddings will fix the concern in the paper that Transformers have GL group symmetry? It is shown in practice that it stabilizes training (such as [1]). [2] also uses spherical projection for theoretical convenience. Or if not, are there alternative ways to address the scaling symmetry?

[1] Scaling rectified flow transformers for high-resolution image synthesis. Esser et al. 2024.
[2] The emergence of clusters in self-attention dynamics. Geshkovski et al. 2024.

---

> ### Author Response · Authors · 2024-11-25
> **Response to Reviewer qb83 - Part 1 of 2**
>
> We thank the reviewer for their feedback and for their suggestions on how to improve the paper. Below we address your questions:
>
> **W1**
> > My main concern is the practical effectiveness of the proposed method: does it really improve existing adaptive sharpness? In Figure 4 the improvements are not quite visible, and the parameter is neither explained nor aligned by column.
>
> We have added Table 1 at the start of the appendix with this information. The parameter tau is kendall tau correlation coefficient, which measures rank correlation of the results: if the sharpness of a model a is larger than that of model b, then, if sharpness is indeed correlated with generalization, we expect the generalization gap of model a to be larger than that of model b if the correlation is positive and negative otherwise.
>
> **W2**
> > The experiments are not described in detail, which affects reproducibility. For self-containedness, the method used in the experiment from Wortsman et al. (2022) could be explained better in detail. For example, what are the hyperparameters, how are the models trained, what is the precise algorithm, etc.?
>
> Thank you for this suggestion. We have added an appendix (Appendix C) with a description of the algorithm we use for solving for geodesic sharpness, which hews closely to that used to solve for adaptive sharpness. In terms of the models from Wortsman (and the new BERT models) we take the weights of the already trained models from the authors. The hyperparameters we use for solving for geodesic sharpness are identical to the ones used by [1]. We hope this provides sufficient clarity, but we are of course happy to provide any additionally required details.
>
> **W3**
> > It is questionable whether the attention's symmetry argued by the author is indeed the practical reason rather than a hypothesis. It would be helpful if the author could give a stronger argument or some illustrations. To give an example, ensure that the scaling symmetry causes irregularity in pre-trained models by measuring the conditional number, and modulating out this symmetry indeed corrects the sharpness---or any other way you prefer.
>
> We have added a small example in Appendix D.1 (Figure 7) extending Figure 1 that we hope will go some way to addressing this. In it, we plot, for our scalar toy model, the level sets of: a) the loss function; b) the euclidean and riemannian gradient; c) the traces of the euclidean and riemannian network hessian. The reason we picked the trace of the network Hessian is that it is a quantity that has been used to quantify flatness. Of note in that figure is that a) the Riemannian version of the gradient and Hessian have the same level set geometry as the loss function; b) both the riemannian gradient norm and the trace of the riemannian hessian have smaller values throughout than their euclidean equivalents; c) the trace of riemannian hessian actually reaches 0 when at the local minimum, whereas the euclidean hessian actually attains its highest value there; d) the Euclidean trace of the hessian is unable to distinguish between a local minimum and a local maximum (upper left vs bottom left in the figure) whereas the Riemannian trace can actually do so, something which is desirable if we want to study generalization.
>
> **W4-5**
>
> Agreed, we have made some changes that we hope improve the presentation of the paper.

---

> ### Author Response · Authors · 2024-11-25
> **Response to Reviewer qb83 - Part 2 of 2**
>
> **Q1**
> >I don't understand the role of the tau variable in the result figures. For example, could you explain how it is chosen, and how it relates to the results?
>
> As we mentioned in W1, tau is the Kendall tau correlation coefficient. We chose it following previous works that used it as a metric to assess correlation between generalization and various proposed measures to predict it, [2], [3], [4]. A Kendal tau of 1.0 indicates perfect rank correlation and a kendal tau of -1.0 indicates perfect anti-correlation. We improve on this measure (increase its magnitude) compared to adaptive sharpness in our results
>
> **Q2**
> > I don't understand why in Figure 3 (left), the average-case adaptive sharpness is claimed to reveal a correlation with the test loss (generalization gap). To me, it looks not obvious, and the dependency is reversed.
>
> The reviewer makes a good point about the strength of the correlation. It is not very high, but a Kendal tau of 0.4 normally indicates a moderately strong correlation between the ranks of the data points. The plot does not look exactly linear, but for the most part higher values of the average-case adaptive sharpness do produce higher values of the test error. The dependency is indeed reversed, which happens in diagonal networks because average-case adaptive sharpness is the $L^1$ norm of the predictor, $||u\odot v||_1$,  which correlates with generalization when the data is sparse in this case, while the maximum-case sharpnesses are (or are close to being) the $L^{\infty}$ norm of the predictor, which anti-correlates.
>
> **Q3**
> > c inconsistently appears in Appendix D. Do the experiments of geodesic sharpness also choose a vector by taking to normalize each parameter tensor? If not how to guarantee fair comparison? In Appendix D is preset, but I don't understand whether the geodesic is invariant over scaling?
>
> Good point. c is a hyperparameter for adaptive sharpness that geodesic sharpness does not require. We included c in the appendix merely to highlight the connection with adaptive sharpness. We expanded/re-wrote Appendix D in a way that hope makes clear what the connection is and the role of c.
>
> **Q4**
> > I wonder if normalizing QK embeddings will fix the concern in the paper that Transformers have GL group symmetry? It is shown in practice that it stabilizes training (such as [1]). [2] also uses spherical projection for theoretical convenience. Or if not, are there alternative ways to address the scaling symmetry?
>
> Interesting question, and we thank the reviewer for bringing our attention to [2], which we were not aware of and seems quite intriguing. On normalizing QK embeddings: they wouldn’t entirely fix it. The attention operation ($XW_Q@W_K^T$) becomes (X LayerNorm($W_Q$) (LayerNorm($W_K)^T X^T)$, which removes issues with scaling symmetry, but any transformation of $W_Q$ and $W_K$ such that  LayerNorm($W_Q$) -> LayerNorm($W_Q$) $G^{-1}$,LayerNorm($W_K$)->LayerNorm($W_K$) $G^T$ would still result in the same attention matrix. In more intuitive terms, transforming a weight matrix by multiplying with an invertible matrix $\in GL_n$ encompasses besides re-scalings ($G = diag(\alpha)$), things (among others) like rotations (for G orthogonal), we expect that the introduction of LayerNorm would fix re-scalings but not necessarily the rotations for example. While we are not entirely familiar with [2] we suspect the same would hold there. We are not aware of any alternative ways of addressing this additional scaling symmetry during training.
>
>
> [1] Andriushchenko et al. (2023) A Modern Look at the Relationship between Sharpness and Generalization
>
> [2] Jiang et al. (2019) Fantastic Generalization Measures and Where to Find Them
>
> [3] Dziugaite et al. (2020). In search of robust measures of generalization
>
> [4] Kwon et al. (2021).  ASAM: Adaptive Sharpness-Aware Minimization for Scale-Invariant Learning of Deep Neural Networks

---

### Official Review · Reviewer_Yq4W · 2024-11-03

**Soundness:** 2
**Presentation:** 2
**Contribution:** 3
**Rating:** 3
**Confidence:** 3

**Summary:**

This paper presents a novel analysis of sharpness in machine learning, with a particular focus on its implications for generalization. The authors argue that existing sharpness measures are inadequate due to their failure to account for the symmetries which arise in parameter space, especially the complex symmetries inherent in transformer architectures. To address this limitation, they leverage tools from Riemannian geometry to propose a refined definition of sharpness (their so-called geodesic sharpness) that explicitly removes these symmetries by instead focusing on the quotient space. This construction draws upon concepts from the theory of quotient manifolds. The authors validate their theoretical framework through experiments on both toy models and vision transformer models trained on ImageNet.

**Strengths:**

I found this to be an interesting paper. The authors explore the notion and role of sharpness in machine learning and shed light on how sharpness is affected by the symmetries which arise in parameter space for these large machine learning models. They use interesting ideas from Riemannian geometry and provide a clear, detailed analysis of difficult ideas. The ideas are original and they provide a creative way and mathematically sound way to define a new notion of sharpness (i.e., geodesic sharpness) which mitigates this issue of symmetry in parameter space.

**Weaknesses:**

Based on my understanding of the author's work, here are the weaknesses I see

There are a few conceptual gaps for me. While the use of quotient spaces to address parameter symmetries is conceptually nice, certain aspects of their construction are unclear to me. In particular, the choice of Riemannian metric on the total space $\bar{\mathcal{M}}$ appears to be crucial, as exemplified by the metric defined in equation (15) for attention layers.  This metric is not unique or canonical. Would similar results hold for any metric, except I suppose for the standard Euclidean one? It is unclear whether the proposed sharpness measure is invariant under different choices. One could conceive of a metric on $\bar{\mathcal{M}}$ that yields a different value for geodesic sharpness. Is that right? Or should all measures for geodesic sharpness be invariant under the R. metric of the total space as well? This technique is marketed in the introduction/conclusion as a "one-size-fits-all" technique for appropriately measuring sharpness but this choice of total R. metric doesn't feel that way to me. To address this concern could the authors
 1. Explain the rationale behind choosing the specific metric in equation (15)
 2. Provide some discuss around how sensitive the results are to different choices of Riemannian metrics on the total space
 3. Clarify if there's a principled way to select an "optimal" metric for a given architecture.
These points would help evaluate whether the method is truly "one-size-fits-all" as claimed.

Also the details of how to actually compute the geodesic sharpness for the attention layer example (Sec 5.2) is lacking, see line 468-469. It's not clear to me what that means to "...plug Eq. 16 into Eq. 17 and solve the resulting optimization problem using SGD". Would it be possible to include the details for that computation and optimization solution details/setup? To address this concern, could the authors
 1. provide the explicit form of the optimization problem after plugging Eq. 16 into Eq. 17 and clarify the specific SGD algorithm used to solve it and hyperparameters used, and
 2. provide any additional constraints or modifications needed to ensure the optimization remains on the manifold

It would be nice to see more experiments or at least some more computational examples. The paper only addresses, in a careful way, very simple diagonal networks and attention layers in transformers. Nothing in between. The experiments are limited to correlational studies. It would be good to have more architectures and datasets to help convince the reader of this new technique. To address this, the authors could
 1. test this measure of geometric sharpness on intermediate architectures between diagonal networks and transformers, such as MLPs or CNNs; or ideally, architectures that are more closely comparable with other measures of sharpness in the literature.
 2. apply the technique to different datasets beyond ImageNet
 3. conduct ablation studies to isolate the impact of different components of the 'geodesic sharpness' measure

Also, the final takeaway message from Figure 4 (see line 488) is that the "...geodesically sharpest models studied on ImageNet are those that generalise best." This seems counter to the typically held belief that less sharp or more flat models generalize better. It would be nice to have more experiments or theory to explain why we see the opposite behavior here than previously in the literature. Or it would be nice to recreate the experiments in the literature which demonstrate that less sharp models generalize better (e.g. the original SAM paper and followups) and show that the geodesic sharpness behaves in the opposite way. To address this concern, could the authors
 1. provide a theoretical explanation or justification as to why 'geodesic sharpness' might behave differently from traditional sharpness measures
 2. conduct a direct comparison with previous sharpness measures on the same models and datasets. This, in particular, would be very nice. And
 3. discuss potential implications of this result for providing better understanding of generalization in deep learning

There are also some parts of the exposition that I found confusing or misleading (in either the concepts or lack of details in plots) and quite a large number of typos which made the paper hard to follow at times. I've outlined these points in the questions section below.

**Questions:**

- line 112. in contribution (d) you claim that Figure 1 shows that there is a strong correlation between generalization and sharpness. Is this a typo? Can you better explain the connection between Figure 1 and generalization?
- In general, the paper appears to only address symmetries that arise as group actions of GL. Is it obvious that this captures all possible symmetries in the parameter space? What is the effect of your construction of the geodesic sharpness when not all symmetries are removed?
- in contribution (f) and in the paper you make a comparison with the CLIP  experiments from Andriushchenko et al. However, those models were not trained to convergence and one could argue that the role of sharpness in generalization is only evident for well-tune and converged models. Does that also apply to your measure of geodesic sharpness?
- line 279, why denote the points in total space by $\bar{x}^{(')}$. I'm confused by this notation.
- line 280, is this a typo? Do you mean x,y for points in the quotient space?
- line 297. Can you clarify the details in the section "Linear embedding space". When you say "On the outer most layer, we are given a linear Euclidean space E…" what does this mean? Do you mean the $d$ parameters relating to the outermost layer of the network? Also, why define the loss $\overline{\overline{\ell}}$ just on these outer layer parameters? How are $\overline{\overline{\ell}}$ and $\overline{\ell}$ related to each other?
- then on line 304, you reference the "Riemannian generalization" The R. generalization of what? The gradient?
- line 318, i mentioned this above, but here you reference endowing the total space $\overline{\mathcal{M}}$ with a smooth inner product. Yes. it is possible to endow any smooth manifold with a Riemannian metric. However, this R. metric is not unique. You reference defining it in Appendix B.4 but that just points to a definition of what a Riemannian metric is. This is confusing and crucial aspect of this work.
- Section 4. When defining the geodesic sharpness, I'm surprised to not see mention of the exponential map to define geodesics in the manifold using vectors in the tangent space. Would it not be possible to reformulate your construction looking only at vectors in the tangent space? And restrict yourself to vectors of a certain size in order to keep them within a fixed ball when projected to the manifold.
- In Figure 3, you plot sharpness vs test loss for diagonal models. But only for 50 of the 200 models from the experiment setup? Why only 50 and why those?
- How natural or reasonable is Assumption 5.1? There is a low-rank bias in deep learning when training overparameterized models. Can you provide a reference that this assumption is usually satisfied in multi-head attention layers for default choices of $d_v, d_k$?
- line 453: Is it possible to endow the total space with any smooth Riemannian metric? Why this one? Admittedly I'm less familiar with the reference (Absil et al 2008) but it would be nice to see a computation verifying this is indeed a Riemannian metric and why this inner product was chosen.
- line 514: Can you clarify what you mean when you say "..attention weights approach being singular" here? How are you defining singular. Also you mention a relaxation parameter but I'm not clear on the role this 'relaxation parameter' plays. You indicate that "..in practise we found out results were robust to this parameter". Can you explain more what robustness analysis you conducted to justify this claim.

typos/nits
 - (typo) Do you mean $L_{\mathbb{S}}$ not $L_{S}$  in equations (1) and (2)? same in equation (3)
 - (nit/personal preference) := is more common that :- for definitional equivalence
 - (typo) line 229. Do you mean $\overline{\overline{f}}$ is symmetric…?
 - the Figures have been resized and the font is very small, difficult to read
 - line 434: what do you mean that the GL(h) symmetry is 'dealt' by geodesic sharpness?
 - line 463: 15 should be (15)

---

> ### Author Response · Authors · 2024-11-25
> **Response to Reviewer Yq4W - Part 1 of 5**
>
> We thank the reviewer for their thorough reading of our paper. We are glad they appreciate the creativity of our ideas and the mathematical soundness of our approach, and we truly appreciate their constructive criticism, which has been helpful in improving the paper. We feel that the thoughtful, detailed level of their suggestions has made it possible for us to fully address almost all of their points, and thus significantly strengthen the paper through doing so.
>
> **W1**
> > 1) Explain the rationale behind choosing the specific metric in equation (15)
> > 2) Provide some discuss around how sensitive the results are to different choices of Riemannian metrics on the total space
> > 3) Clarify if there's a principled way to select an "optimal" metric for a given architecture. These points would help evaluate whether the method is truly "one-size-fits-all" as claimed.
>
> 1) **Rationale for choice of metric**. Thank you for raising this important question. There do indeed exist multiple choices: this is a general degree of freedom in Riemannian geometry, and it is a priori not clear which metric will yield the best results (e.g. there are natural gradient descent flavors based on the Fisher-Rao or the Wasserstein metric; both are used in practice). In our case we don’t have complete freedom to pick any metric we want: whatever metric we use has to be compatible with the symmetry we are trying to factor out. This requirement immensely restricts the metrics we can pick, but still does not result in a single unique possible choice. The reasons we chose the specific metric that we did are two-fold (in order of importance): a)  this metric reduces nicely in the case of re-scaling symmetry - as those of the diagonal networks or adaptive sharpness -,  to quantities known to correlate with generalization, providing a signal on the suitability of this particular metric; b) the quotient space geometry was already characterized by Absil et al.
>
> 2) **Sensitivity to different metrics** Ignoring the trivial cases of metrics that are related to metric (15) via re-scalings (as these result in the same geodesic sharpness, see appendix I.3) , we are aware of one other metric that yields a quotient geometry that has previously been characterized. In Appendix F.2 we describe this alternative metric. In our initial experiments the results did not substantially change.
>
> 3) **Principled approaches to metric selection** We are not aware of any principled way of selecting this metric, besides making sure it reduces to a useful measure in the usual settings (such as the scalar re-scalings of adaptive sharpness).
>
> **W2**
> > Also the details of how to actually compute the geodesic sharpness for the attention layer example (Sec 5.2) is lacking, see line 468-469. It's not clear to me what that means to "...plug Eq. 16 into Eq. 17 and solve the resulting optimization problem using SGD". Would it be possible to include the details for that computation and optimization solution details/setup? To address this concern, could the authors
> > 1. provide the explicit form of the optimization problem after plugging Eq. 16 into Eq. 17 and clarify the specific SGD algorithm used to solve it and hyperparameters used, and
> > 2. provide any additional constraints or modifications needed to ensure the optimization remains on the manifold
>
> Thank you for pointing this out; we agree that this section was lacking detail. We have added an appendix (appendix C) to the paper explaining in detail how the computation works and have also included some pseudo-code. We hope that this clarifies the paper in this regard.

---

> ### Author Response · Authors · 2024-11-25
> **Response to Reviewer Yq4W - Part 2 of 5**
>
> **W3**
> > It would be nice to see more experiments or at least some more computational examples. The paper only addresses, in a careful way, very simple diagonal networks and attention layers in transformers. Nothing in between. The experiments are limited to correlational studies. It would be good to have more architectures and datasets to help convince the reader of this new technique. To address this, the authors could
> > 1. test this measure of geometric sharpness on intermediate architectures between diagonal networks and transformers, such as MLPs or CNNs; or ideally, architectures that are more closely comparable with other measures of sharpness in the literature.
> > 2. apply the technique to different datasets beyond ImageNet
> > 3. conduct ablation studies to isolate the impact of different components of the 'geodesic sharpness' measure
>
> First, we strongly agree with the reviewer’s overall point of providing additional empirical evidence (which we did, with results further supporting our approach), and we appreciate the reviewer’s depth of engagement to suggest three different approaches to doing so, and we address each one individually):
>
> 1. **Intermediate architectures.** This is an interesting idea, but the tricky part is determining what would be an appropriate “intermediate” architecture. We considered MLPs and CNNs, but, while it is not immediately obvious, we suspect that results would be quite similar to those that would be obtained by adaptive sharpness in those settings. The reason is that in this case,  the $GL_N$ re-scaling symmetry reduces to a $GL_1+$ symmetry, and in that setting geodesic sharpness almost fully reduces to the usual adaptive sharpness (we still have the symmetry curvature term). We have added a new appendix, Appendix F.1 in which we show this explicitly in a new appendix  (and which at the same time we hope also provides additional evidence for the choice of metric in the paper). As (admittedly merely suggestive) evidence of this being the case, we do note that in the case of diagonal networks where this reduction happens, geodesic sharpness mirrors adaptive sharpness: it does have a slightly higher tau correlation factor but for the most part it essentially behaves as adaptive sharpness.
>
> 2. **Datasets beyond Imagenet.** This was indeed one of our top priorities, and in the revision we have included the results of geodesic sharpness for a pre-trained BERT on MNLI in the paper. Our approach again clearly improves the result obtained over using adaptive sharpness.
> 3. **Ablating for different components of the geodesic measure**.  Thank you for this good suggestion; we will add the full results of this to our paper in an appendix, but in our preliminary experiments, the new norm impacts results the most, but the adding the second order weight corrections do not matter as much. Both components are necessary for the best performance. Details can be found in Appendix G

---

> ### Author Response · Authors · 2024-11-25
> **Response to Reviewer Yq4W - Part 3 of 5**
>
> **W4**
>
> > Also, the final takeaway message from Figure 4 (see line 488) is that the "...geodesically sharpest models studied on ImageNet are those that generalise best." This seems counter to the typically held belief that less sharp or more flat models generalize better. It would be nice to have more experiments or theory to explain why we see the opposite behavior here than previously in the literature. Or it would be nice to recreate the experiments in the literature which demonstrate that less sharp models generalize better (e.g. the original SAM paper and followups) and show that the geodesic sharpness behaves in the opposite way. To address this concern, could the authors
> > 1. provide a theoretical explanation or justification as to why 'geodesic sharpness' might behave differently from traditional sharpness measures
> > 2. conduct a direct comparison with previous sharpness measures on the same models and datasets. This, in particular, would be very nice.
> > 3. And discuss potential implications of this result for providing better understanding of generalization in deep learning
>
> This is an excellent question that we have been thinking about (and recently excited about, as we indicated in the general response to all reviewers).
> 1. **Theoretical Explanation** While we do not have a fully fleshed out theoretical explanation to offer, we can hope that what follows at least provides some justification. In diagonal networks when we train on sparse data, the $L^1$ norm of the predictor $u \odot v$ is a good predictor of generalization, whereas when we train on dense data, the $L^{\infty}$ norm tends to be more useful. Maximum adaptive sharpness (as compared to average adaptive sharpness) with a specific hyperparameter choice, becomes $||u \odot v||_{\infty}$ and so is positively correlated when the networks are trained on dense data, and negatively so when trained on sparse data. The reverse happens for average adaptive sharpness. We speculate something similar might be happening here, but both the architecture and the nature of the data itself is considerably more complex and so we cannot, yet, fully understand it. Teasing out the exact nature of these interactions is definitely something we intend to explore, and we feel that doing so properly will involve systematic investigation along several axes, so to speak, and this will be beyond the scope of this current paper. We are very excited to explore this in future work.
> 2. **Comparison with previous sharpness measures (e.g. SAM)** This is an intriguing idea. As mentioned in W.3 for CNNs and MLPs we don’t expect much of a difference with conventional adaptive sharpness, and so the experiments in the original SAM paper will probably not be particularly enlightening. Does the reviewer have a specific SAM followup paper dealing more with transformers in mind? We would be happy to try to reproduce as much as we can.
> 3. **Potential implications of this result**. Indeed, potential implications for better understanding generalization is what we are very much interested in. Our view is that this result,  as it stands, an essential first step in this direction. We carefully introduced a new metric, and this metric allowed us to recover a previously elusive correlation between sharpness and generalization. Doing so has already opened  exciting new questions around it, the sign of the correlation being one of them. Our perspective is that if we want to understand more deeply how sharpness relates to generalization, we at the very least needed to have a notion of sharpness that is useful for transformers and that actually does correlate with generalization. In this work, we have empirically verified this. Building on this in future, we hope (speculate!) that the general approach we used to produce this notion of sharpness: a) provides a more intrinsic view of the parameter space, allowing for access to objects such as the Riemannian curvature and Hessian on the quotient manifold, which we haven’t touched in this paper and might yet prove useful; and b) will help us to fold in more general symmetries and possibly the role of data into this quotient framework, which in turn might allow us to further refine what sharpness ought to mean.

---

> ### Author Response · Authors · 2024-11-25
> **Response to Reviewer Yq4W - Part 4 of 5**
>
> **Q1**
> > line 112. in contribution (d) you claim that Figure 1 shows that there is a strong correlation between generalization and sharpness. Is this a typo? Can you better explain the connection between Figure 1 and generalization?
>
> This was definitely a typo, thank you for pointing it out.
>
> **Q2**
> > In general, the paper appears to only address symmetries that arise as group actions of GL. Is it obvious that this captures all possible symmetries in the parameter space? What is the effect of your construction of the geodesic sharpness when not all symmetries are removed?
>
> The reviewer is completely right that we are not accounting for all possible symmetries. For instance permutation symmetry, exploited to great effect in [1], is not fully encompassed in our approach since we use a more restricted re-scaling symmetry for the non-attention parameters. It could conceivably be incorporated since permutation matrices are invertible, but we have not done so yet. But besides these there might still be other, more complex symmetries that we do not account for. Our view is that it is better to at least account for some (as shown in the improved results) than none, but we hope to extend this in the future.
>
> **Q3**
> > in contribution (f) and in the paper you make a comparison with the CLIP experiments from Andriushchenko et al. However, those models were not trained to convergence and one could argue that the role of sharpness in generalization is only evident for well-tune and converged models. Does that also apply to your measure of geodesic sharpness?
>
> We suspect it does, but in the same way that sharpness is useful during training (for SAM, for example), we think our measure could have the same use in training.
>
> **Q4**
> > line 279, why denote the points in total space by I'm confused by this notation.
> Thank you for pointing this out. It is a typo, as the prime should not be there. We have fixed this in the paper.
>
> **Q5**
> > line 280, is this a typo? Do you mean x,y for points in the quotient space?
>
> Thank you for pointing this out. We have fixed this in the paper.
>
> **Q6**
> > line 297. Can you clarify the details in the section "Linear embedding space". When you say "On the outer most layer, we are given a linear Euclidean space E…" what does this mean? Do you mean the parameters relating to the outermost layer of the network? Also, why define the loss just on these outer layer parameters? How are and related to each other?
>
> This section was not well written in the paper and the mention of the outer layer is an oversight. We changed it, but in summary: our parameters ultimately live on computers and in principle can be any real matrix, this we call our embedding space; we are, by assumption 5.1 working with full-rank matrices, so our total space is that of full-rank matrices and from this total space we construct the quotient manifold.
>
> **Q7.**
> > parameters relating to the outermost layer of the network? Also, why define the loss just on these outer layer parameters? How are and related to each other?
>
> We would answer this roughly in the same way as in Q6. The loss is defined on the network parameters. We have re-worked this section in the paper.
>
> **Q8**
> > line 318, i mentioned this above, but here you reference endowing the total space with a smooth inner product. Yes. it is possible to endow any smooth manifold with a Riemannian metric. However, this R. metric is not unique. You reference defining it in Appendix B.4 but that just points to a definition of what a Riemannian metric is. This is confusing and crucial aspect of this work.
>
> We hope the discussion in W1 served to address this.
>
> **Q9**
> > Section 4. When defining the geodesic sharpness, I'm surprised to not see mention of the exponential map to define geodesics in the manifold using vectors in the tangent space. Would it not be possible to reformulate your construction looking only at vectors in the tangent space? And restrict yourself to vectors of a certain size in order to keep them within a fixed ball when projected to the manifold.
>
> Indeed it would. In equation 7, $\gamma_{\bar{\xi}}$(1) is just the exponential map at a horizontal vector $\bar{\xi}$. The reason we omitted this from the paper is that we tried to make the paper as accessible as possible to readers without more than cursory knowledge of riemannian geometry, and so strived to introduce the smallest possible set of concepts from it in the paper. We are open to changing this if the reviewer feels the presentation would gain from doing so.
>
> **Q10**
> > In Figure 3, you plot sharpness vs test loss for diagonal models. But only for 50 of the 200 models from the experiment setup? Why only 50 and why those?
>
> This is an oversight for which we apologize: we trained 50 networks and then showed all of them.

---

> ### Author Response · Authors · 2024-11-25
> **Response to Reviewer Yq4W - Part 5 of 5**
>
> Q11
> > How natural or reasonable is Assumption 5.1? There is a low-rank bias in deep learning when training overparameterized models. Can you provide a reference that this assumption is usually satisfied in multi-head attention layers for default choices of dv, dk
>
> In general in non-linear networks there is indeed a tendency towards low-rank representations, which might make Assumption 5.1 seem excessive and counter to realistic situations. However, while the learned $W_Q W_K^T$ tends to be low-rank, $W_Q$ and $W_K$ (on which Assumption 5.1 ought to apply) themselves are usually high/full (column) rank [2].  In practice, this is also the case for the models that we study in this paper.
>
> **Q12**
> > line 453: Is it possible to endow the total space with any smooth Riemannian metric? Why this one? Admittedly I'm less familiar with the reference (Absil et al 2008) but it would be nice to see a computation verifying this is indeed a Riemannian metric and why this inner product was chosen.
>
> Added in appendix I.2 a computation showing it is indeed a Riemannian metric. Hope the first part is addressed by our answer to W1.
>
> **Q13**
> > line 514: Can you clarify what you mean when you say "..attention weights approach being singular" here? How are you defining singular. Also you mention a relaxation parameter but I'm not clear on the role this 'relaxation parameter' plays. You indicate that "..in practise we found out results were robust to this parameter". Can you explain more what robustness analysis you conducted to justify this claim.
>
> We here mean singular to mean $W_{Q,K,V,O}$ are not full-rank. Since we need to invert matrices of the type of $W_Q^T W_Q$ this can create a numerical issue. Because of rounding errors of floating-point precision, in practice $W_Q^TW_Q$ is always invertible, but sometimes the inverted matrices have large singular values. To combat this we introduce the relaxation parameter to $W_Q^TW_Q$, which dampens the resulting singular values. While we can’t take it to be exactly zero, as long as it’s small enough, numerical stability is improved and the results stay roughly the same (we show some results of the effect of varying this parameter in Appendix H.2 and Figure 11).
>
> **typos/nits**
>
> We have fixed all of these in the paper. We apologize for the size of the plots, they are the way they are due to page limits. We will endeavour to tighten up the presentation to find space for them. We have removed the “dealt by […] line in question as it does not add anything.
>
> [1] Samuel K. Ainsworth et al. (2022). Git Re-Basin: Merging Models modulo Permutation Symmetries
>
> [2] Yu, H., & Wu, J. (2023). Compressing Transformers: Features Are Low-Rank, but Weights Are Not!

---

### Official Review · Reviewer_KB1X · 2024-11-04

**Soundness:** 3
**Presentation:** 3
**Contribution:** 3
**Rating:** 8
**Confidence:** 4

**Summary:**

This work introduces a new approach to measuring model sharpness using Riemannian geometry. This approach takes into account the symmetries in model parameters, removing ambiguities that these symmetries can cause. As a result, it provides a clearer and more reliable sharpness measure compared to previous methods.

**Strengths:**

1. This work takes a broader view of parameter symmetries and introduces a more robust metric for measuring sharpness.

2. The theoretical framework and assumptions are well-grounded, and the experimental results are both strong and consistent.

**Weaknesses:**

1. $\textbf{Geodesic sharpness}$ in section 5.2 is abbreviated. Providing a clear, explicit formula for geodesic sharpness would improve comprehension.

2. The discussion of geodesic sharpness in transformers (Section 5.3) is brief. Including a description of geodesic sharpness for transformers, or a brief explanation of how adaptive sharpness could be applied to each layer, would make the analysis more thorough.

3. The paper aims to propose a metric that surpasses previous methods. Therefore, I would expect comprehensive experimental results demonstrating both the effectiveness of the proposed approach and its superiority over existing methods. While the experiments provide sufficient evidence of the method’s effectiveness, the comparative analysis with previous methods seems limited.

Minor points:

$\bullet$ Example 3.2 "(Self-attention Vaswani et al. (2017))" $\to$ "Example 3.2 (Self-attention (Vaswani et al., 2017))".

$\bullet$ $\textbf{Geodesics}$ in section 5.2: "... the geodesics of metric 15" $\to$ "... the geodesics of metric (15)".

**Questions:**

1. In Case A), the second-order term in Equation (12) is omitted because it is considered small compared to the first-order term. However, I wonder if this omission might be overly convenient. Since there is no indication of how small $\rho$ can be, I’m unsure about the relative scale of the first and second orders. Would it be possible to retain the second-order term and, perhaps, combine the results of both cases to reach the conclusion?

2. Could you provide further clarification on why "adaptive sharpness is more appropriate" in Section 5.3?

---

> ### Author Response · Authors · 2024-11-25
> **Response to Reviewer KB1X**
>
> **W1**
> > Geodesic sharpness in section 5.2 is abbreviated. Providing a clear, explicit formula for geodesic sharpness would improve comprehension.
>
> We have changed this in the paper, and we hope it now reads a bit more clearly. But we can make further adjustments if needed.
>
> **W2**
> > The discussion of geodesic sharpness in transformers (Section 5.3) is brief. Including a description of geodesic sharpness for transformers, or a brief explanation of how adaptive sharpness could be applied to each layer, would make the analysis more thorough.
>
> We added this as  appendix C.1 in the paper, as we agree it was unclear in the previous version. In summary: we treat the ($W_Q$, $W_K$), and ($W_V$,$W_O$) of each head ($W_O$ requires a bit more care) as $GL_h$ symmetric pairs, and treat all other parameters as having the same symmetry as is accounted for in adaptive sharpness, that of element wise re-scalings. We then follow Algorithm 1 in Appendix C.2 (essentially projected gradient ascent) to obtain the maximum sharpness.
>
> **W3**
> > The paper aims to propose a metric that surpasses previous methods. Therefore, I would expect comprehensive experimental results demonstrating both the effectiveness of the proposed approach and its superiority over existing methods. While the experiments provide sufficient evidence of the method’s effectiveness, the comparative analysis with previous methods seems limited.
>
>  We have included additional experiments on real-world language data, and again observed an increase in the correlation compared with adaptive sharpness. Encouragingly, we managed to improve a tau of roughly zero (from adaptive sharpness), indicating no correlation at all, to a tau with a clear correlation, and with visible order.
>
> **Minor points**
>
> We have fixed these issues in the paper.
>
> **Q1**
>
> > In Case A), the second-order term in Equation (12) is omitted because it is considered small compared to the first-order term. However, I wonder if this omission might be overly convenient. Since there is no indication of how small can be, I’m unsure about the relative scale of the first and second orders. Would it be possible to retain the second-order term and, perhaps, combine the results of both cases to reach the conclusion?
>
> That is a good point, yes, it would. We added this analysis to Appendix E. Depending on the residual and rho, the results don’t necessarily change much.
>
> **Q2**
> > Could you provide further clarification on why "adaptive sharpness is more appropriate" in Section 5.3?
>
> That section of the paper was slightly re-written to be clearer, and we expand on it in Appendix C.1. In essence however, convolutional or fully connected layers do not have the full GL symmetry of the attention parameters, having only a re-scale symmetry, which is the symmetry which adaptive sharpness accounts for.

---

### Official Review · Reviewer_VDTG · 2024-11-04

**Soundness:** 3
**Presentation:** 2
**Contribution:** 3
**Rating:** 6
**Confidence:** 3

**Summary:**

This paper proposes a method for measuring the sharpness of neural networks, called geodesic sharpness, which takes into account the symmetries present in the network architecture. The authors argue that previous sharpness measures fail to accurately capture the sharpness of transformers because they do not account for the rich symmetries present in the attention mechanism. The authors clearly motivate the need for a new sharpness measure that is invariant to these symmetries. However, the paper need to be polished more to fix typo and clearer explanation, especially experiments.

**Strengths:**

From theoretical aspects, the authors' main theoretical contribution is the application of Riemannian geometry to the study of neural network parameter space symmetry. They propose to use the geometry of the quotient manifold, which is obtained by removing the symmetries from the parameter space, to define a sharpness measure that is invariant to these symmetries. This approach is general and can be applied to a wide range of symmetries. The authors show that ignoring the curvature introduced by the symmetries leads to traditional adaptive sharpness measures.

From experiment aspects, yo validate their approach, the authors conduct experiments on both synthetic and real-world data. They analytically derive the geodesic sharpness for diagonal networks and show that it correlates strongly with generalization. They also apply their method to large vision transformers and find that geodesic sharpness has a stronger correlation with generalization than any previously reported measure, both for in-distribution and out-of-distribution settings.

The main strength of the paper is its novel and principled approach with the use of Riemannian geometry to account for symmetries in sharpness measures, which can apply to more problem like equivariant neural functional network.

**Weaknesses:**

The paper, while presenting a promising theoretical approach to measuring sharpness, suffers from certain shortcomings related to the experimental validation of the proposed method and the potential computational cost involved:

-   **Limited empirical support:** The experiments conducted on real-world transformers lack breadth.It would be good if the authors can demonstrated in the natural language task too, like language modeling with wikitext103.

-   **Effect of assumptions:** The paper makes an assumption regarding the full column rank of weight matrices in attention layers. A relaxation parameter is introduced to handle potential violations of this assumption, but the impact of this parameter on the results is not thoroughly investigated.

-   **Computational burden:** The paper acknowledges the potential computational demands of calculating geodesic sharpness, especially for large models. However, it lacks a detailed analysis of the computational cost, including aspects like time complexity, memory requirements, and actual runtime measurements. Such an analysis would enable a better understanding of the feasibility of employing geodesic sharpness in practical scenarios, particularly when dealing with large-scale models.

Addressing these weaknesses would bolster the paper's contributions and enhance the understanding of the interplay between sharpness and generalization in the context of transformer models. I willing to increase the score more if the authors can provide more convincing analysis on these weaknesses.

**Questions:**

1. "RIEMMANIAN" -> "RIEMANNIAN" in title
2. ",and" -> ", and" in line 39
3.  In my opinion, the contributions can be collated, where (b)-(c) and (d)-(e)-(f) are collated into two contributions: the development of geodesic sharpness and experiments for it.
4. Diagonal network experiment:  it is not clear to me what Figure 3 is showing. What is the meaning of the x axis and y axis? What is the ideal result? Why only show 50 data points when you have 200 model trained? Why the points in adaptive average case aligned in different direction in comparison with two other cases? I suggest adding more detailed captions and axis labels to Figure 3, and include explanations for these points in the corresponding text.
5.  Assumption 5.1: Does Assumption 5.1 are too strong? What if the assumption is violated? Can we still calculated the result but with wrong value, or not able to calculate it at all? It would be good if the author can provide the evident and explanation of effect of adding $\epsilon I_h$ into $GH$ since there have not any for the assumption. Also, including a discussion of the implications of violating Assumption 5.1 and provide empirical evidence for the effects of adding the $\epsilon I_h$ term.
6. Can the authors describe in detail how you treat the multi-layer schema of transformers model? It is not clear at this time. It would be good if the author can provide the diagram illustrating the approach for multi-layer transformers.
7. Experiment diversity: it would be good if we can demonstrated in the natural language task too, like language modeling with wikitext103, to ensure that it works across domains.
8.  In order to find the geodesic approximation, you need to solve the optimization using SGD, what is the quantitative performance, i.e. wall clock time or time/memory complexity of this approach in comparison with adaptive sharpness? How does it scale with large models like LLMs?

---

> ### Author Response · Authors · 2024-11-25
> **Response to Reviewer VDTG - Part 1 of 2**
>
> We thank the reviewer for taking the time to review our paper, for their overall positive response, their very concrete suggestions on how to strengthen the paper, and their willingness to reconsider their score.
>  1) **Limited empirical support**: we have included additional experiments on real-world language data, and again observed an increase in the correlation compared with adaptive sharpness. Encouragingly, we managed to improve a tau of roughly zero (from adaptive sharpness), indicating no correlation at all, to a tau with a clear correlation, and with visible order.
> 2) **Effect of assumptions**:  we discuss this in more detail below, but in summary: while the multiplied together attention weights tend to be low rank, the individual attention weights (on which we need assumption 5.1 to hold) tend to be high/full-rank. In appendix H.2 we provide evidence that the relaxation parameter does not impact the final results.
> 3) **Computational burden**: we have provided a complexity analysis of the proposed method [Appendix C for details]; essentially showing that our approach is the same order of computation time as adaptive sharpness.
>
> In the next comment we address your detailed questions.

---

> ### Author Response · Authors · 2024-11-25
> **Response to Reviewer VDTG - Part 2 of 2**
>
> Q1-3: These changes are now in the revised paper.
>
> Q4:
> > Diagonal network experiment: it is not clear to me what Figure 3 is showing. What is the meaning of the x axis and y axis? What is the ideal result? Why only show 50 data points when you have 200 model trained? What is the meaning of the Kendall Tau? Why the points in adaptive average case aligned in different direction in comparison with two other cases? I suggest adding more detailed captions and axis labels to Figure 3, and include explanations for these points in the corresponding text.
>
> We have added these points to the main paper. On the issue of 50 vs 200 models, that was a typo; we apologize and we have fixed this. The x axis are the various sharpness measures, the y axis the test loss. The parameter $\tau$ is kendall tau correlation coeficient, which measures rank correlation of the results: if the sharpness of a model a is larger than that of model b, then, if sharpness is indeed correlated with generalization, we expect the generalization gap of model a to be larger than that of model b if the correlation is positive and negative otherwise. The reason for the difference in sign is already present in previous works, flipping depending on the data regime (dense vs sparse data).
>
> Q5:
> > Assumption 5.1: Does Assumption 5.1 are too strong? What if the assumption is violated? Can we still calculated the result but with wrong value, or not able to calculate it at all? It would be good if the author can provide the evident and explanation of effect of adding into since there have not any for the assumption. Also, including a discussion of the implications of violating Assumption 5.1 and provide empirical evidence for the effects of adding the term.
>
> In general in non-linear networks there is a tendency towards low-rank representations, which might make Assumption 5.1 seem excessive and counter to realistic situations. However, while the learned $W_Q  W_K^T$ tends to be low-rank, $W_Q$ and $W_K$ (on which Assumption 5.1 ought to apply) themselves are usually high/full (column) rank [1]. In practice, this is also the case for the models that we study in this paper. Since we need to invert matrices of the type of $W_Q^T W_Q$, not being full-rank can create numerical issues. Because of rounding errors of floating-point precision, in practice $W_Q^TW_Q$ is always invertible, but sometimes the inverted matrices have large singular values. To combat this we introduce the relaxation parameter to $W_Q^TW_Q$, which provably dampens the resulting singular values. While we can’t take it to be exactly zero, as long as it’s small enough, numerical stability is improved and the results stay roughly the same (we show some results of the effect of varying this parameter in Appendix H.2 and Figure 11).
>
> [1] Yu, H., & Wu, J. (2023). Compressing Transformers: Features Are Low-Rank, but Weights Are Not!
>
> Q6:
> > Can the authors describe in detail how you treat the multi-layer schema of transformers model? It is not clear at this time. It would be good if the author can provide the diagram illustrating the approach for multi-layer transformers.
>
> We have added discussion on this to the paper in Appendix C.1, but to briefly summarize here: we treat the $(W_Q, W_K)$, and $(W_V,W_O)$ of each head ($W_O$ requires a bit more care) as $GL_h$ symmetric pairs, and treat all other parameters as having the same symmetry as is accounted for in adaptive sharpness, that of element wise re-scalings. We then follow Algorithm 1 in Appendix C.2 (essentially projected gradient ascent) to obtain the maximum sharpness.
>
> Q7:
> > Experiment diversity: it would be good if we can demonstrated in the natural language task too, like language modeling with wikitext103, to ensure that it works across domains.
>
> We agree, and we have added some experiments with BERT to the paper. They are not on wikitext103, but we chose them because a) there was an adaptive sharpness baseline available; b) there were a large number of these fine-trained models available from the same source. Again we observe an increase in correlation
>
> Q8:
> > In order to find the geodesic approximation, you need to solve the optimization using SGD, what is the quantitative performance, i.e. wall clock time or time/memory complexity of this approach in comparison with adaptive sharpness? How does it scale with large models like LLMs?
>
> Added an appendix (Appendix C.3) discussing this, but in short it’s not much more expensive than adaptive sharpness. We require the same amount of forward and backwards passes, which are in practice the main bottleneck. The main difference is that we require some additional matrix multiplications, but we cache as much as is possible to speed this up. The projection onto the horizontal space requires matrix inversions and can be a bit more expensive, but the matrices that are being inverted are d_head*d_head. We haven’t experimented with models that are larger than around 110M parameters.

---

> > ### Comment · Reviewer_VDTG · 2024-12-03
> >
> > I would like to thank the authors for addressing my concerns in the rebuttal. Overall, all of the answers are fairly satisfactory. Currently, I think the fact that the sharpness and generalization correlation varies across tasks (even varying with different signs) will limit the real applications of geodesic sharpness. Also, the paper still has some typos. However, I appreciate the idea of taking symmetry into account for symmetry-sensitive problems like sharpness and using mathematical tools like differential geometry to solve it. While the experiment results are not very impressive and are somewhat small scale in comparison with the current state of deep learning, the theory provided in the paper will contribute fairly well to further research in this direction. As a result, I would like to raise my score to 6.

---

> > > ### Author Response · Authors · 2024-12-03
> > >
> > > We would like to thank you again for your review and helpful suggestions on how to improve the paper, both on a conceptual and empirical level. The remaining typos will be fixed thoroughly over several passes for any final camera ready version.
> > >
> > > We agree that at the moment our primary contribution is theoretical. That has indeed been the intention of our scope at this point, and our experiments are intended to support our theoretical insight. We are excited because this work opens new possibilities for future exploration incorporating other factors in understanding the relationship of sharpness to generalization. We believe this will also increase practical significance and impact in the long-term.
> > >
> > > We also thank you for carefully considering our response and increasing your score in support of acceptance.

---

### Official Review · Reviewer_tcKK · 2024-11-04

**Soundness:** 4
**Presentation:** 3
**Contribution:** 3
**Rating:** 6
**Confidence:** 3

**Summary:**

This paper proposes a more general notion of sharpness measurement based on a geodesic ball on the symmetry-corrected quotient manifold, which accounts for the symmetry equivalence of the original network. Experiments with Diagonal Networks on synthetic data and transformers on the ImageNet dataset show that Geodesic Sharpness has a stronger correlation with the model’s generalization performance compared to Adaptive Sharpness.

**Strengths:**

- The paper introduces a novel approach to sharpness by leveraging quotient manifolds, reflecting the symmetry of the parameter space. This effectively addresses a gap in previous work that has not considered network symmetry.
- The methodology is theoretically well-founded, employing tools from Riemannian geometry to incorporate network symmetry to the sharpness measure in a principled way.
- Geodesic sharpness is shown to achieve a higher Kendall Tau correlation with the relevant metrics than Adaptive sharpness in both synthetic and ImageNet experiments.

**Weaknesses:**

The experimental section should be strengthened. Assessing the correlation between the proposed sharpness and the generalization of transformers trained on other tasks, such as time series forecasting or language modeling, would further support the findings.

**Questions:**

- Could you include additional experiments on real-world data beyond vision transformers? This would enhance the experiment effectiveness and demonstrate the predictive capacity of the proposed Geodesic sharpness across a broader range of tasks.
- Could you provide a time complexity analysis of the proposed method relative to Adaptive sharpness?

---

> ### Author Response · Authors · 2024-11-25
> **Response to Reviewer tcKK**
>
> We thank the reviewer for their helpful and concrete suggestions. We agree with these suggestions and have carried them out:
>
> - we have included additional experiments on real-world language data, and again observed an increase in the correlation compared with adaptive sharpness. Encouragingly, we managed to improve a tau of roughly zero (from adaptive sharpness), indicating no correlation at all, to a tau with a clear correlation, and with visible structure; and
> - we have provided a complexity analysis of the proposed method [Appendix C for details]; essentially showing that our approach is the same order of computation time as adaptive sharpness.
> Both of these are mentioned in the global response and also discussed in more detail in the paper.
>
> We would be happy to further discuss any other questions.

---

> > ### Comment · Reviewer_tcKK · 2024-12-03
> >
> > Thank you for addressing my concerns about adding new experiments on language task and provide time complexity comparision. Hence I improve my score to 6.

---

> > > ### Author Response · Authors · 2024-12-03
> > >
> > > We would like to thank you again for your review and insightful suggestions to improve the paper.
> > >
> > > We also thank you for carefully considering our response, acknowledging that your concerns are addressed and increasing your original score in support of acceptance.

---

### Author Response · Authors · 2024-12-02
**Summary of Rebuttal and Discussion Period**

As the discussion period draws to a close, we once again thank all the reviewers for their thoughtful comments. We are happy to see that the reviewers unanimously appreciated the originality of our idea (tcKK, VDTG, KB1X, Yq4W, qb83); the principled nature and theoretical richness and soundness of our approach (tcKK, VDTG, KB1X, Yq4W, qb83); and the generality of our approach (VDTG, KB1X,qb83).
The most common recommendations were to add additional experimental results (especially on language models) and discuss optimization and complexity: we have provided both of these in detail, along with other important additions (discussion and experiments on choice of metric, and ablations, for example). In all cases, the new results and analyses support our proposed approach.

We therefore believe we have addressed all of the reviewers’ concerns; for your convenience, we summarize below the new sections and appendices that we added in response to reviewers’ suggestions, along with corresponding main changes made during discussion period and their location in the paper.

**New results for language models (all reviewers)**:[__Section 5.3.2__] We added a section to the paper with results obtained using the fine-tuned BERT models setup from Andriushchenko et al. (2023). We again recover a stronger correlation with generalization than that observed with adaptive sharpness alone.

**Optimization and complexity (tcKK, VDTG, Yq4W)**: [__Appendix C__]  Several reviewers mentioned that a more detailed discussion of the optimization method and its computational complexity was missing. This is now included in an appendix in the paper, Appendix C, showing the feasibility of our approach.

**Illustration of the role of symmetry (qb83)**: [__Appendix D.1__] One reviewer suggested the paper could benefit from an example illustrating the role of symmetry in sharpness and its relationship with generalization. We added a small example (__Figure 7__) extending Figure 1, showing how in a simple setting accounting for parameter symmetry regularizes to some extent the loss landscape and provides a clearer signal as to the generalizing performance of a network.

**Completing the diagonal network analysis (KB1X)**: [__Appendix E__] An appendix with the analysis of geodesic sharpness in diagonal network in the fullest generality was added.

**Reduction of geodesic sharpness to adaptive sharpness and different choices for metrics (qb83, Yq4W)**: [__Appendix F__] The previous discussion on how and when geodesic sharpness reduces to adaptive sharpness was rightly pointed out as being somewhat confusing. This discussion was reworked and expanded and can now be found in [__Appendix F.1__].
One reviewer mentioned that there was some ambiguity in the choice of metric. While the symmetry compatibility criterion restricts the possible choices of metrics immensely, it does not uniquely determine it. We conducted additional experiments with the one other symmetry-compatible metric we are aware off that has commonly been used in the literature. This discussion can be found in [__Appendix F.2__]. This remained consistent with our overall results.

**Ablation studies (Yq4W)**: [__Appendix G__] We added an appendix with ablation studies investigating (and subsequently demonstrating) the importance of both the second-order weight corrections (given by the Christoffel symbols), and the use of the symmetry-compatible norm.


**Assumption 5.1 and the relaxation hyperparameter (VDTG,Yq4W)**: [__Appendix H__] Questions were raised both about the naturalness of assumption 5.1 and the impact that the relaxation parameter has. [__Appendix H.1__] discusses the former, and [__Appendix H.2__] shows the impact the relaxation parameter has on results. Overall, assumption 5.1 is reasonable, and the relaxation parameter does not impact results significantly, while improving numerical stability.

We believe that this addresses all raised concerns by the reviewers (thank you to all of them for their insightful feedback!), and, in sum, it demonstrates the effectiveness, feasibility, and significance of our novel approach to measuring sharpness.

PS. Since writing this summary, two reviewers (VDTG,tckk) have acknowledged our rebuttals so far and subsequently raised their scores. Since all reviewers shared the same set of 2 primary concerns (experiments and computational feasibility), and since (VDTG, tckk) responded that we did satisfy those primary issues and subsequently raised their scores, we very much hope that we have indeed satisfied these concerns for all of the reviewers, along with our other rebuttal points.

---

### Meta-Review · Area_Chair_Lc3z · 2024-12-21

**Metareview:**

This paper proposes to use Riemannian geometry to measure the sharpness of neural networks which explicitly accounts for some of the symmetries in the network by measuring sharpness in the quotient space.  The authors then show that in certain settings this measure of sharpness correlates with model generalization.  The reviewers largely found the concept of the paper interesting and appreciated the theoretical development.  However, the reviewers also noted several weaknesses.  In particular the the reviewers asked for more extensive experimental evaluation, which the authors responded to by adding further experiments on language models and ablation studies.  Along with this there were questions raised by several reviewers regarding the implications of the experimental results.  Notably, the correlation between the sharpness measure and model generalization is often negative (thought the sign of the correlation also appears to potentially switch between experiments), which is counter to the typical intuition for the motivation of studying sharpness.

Overall, the majority of the reviews are somewhat borderline along with one additional positive review and one additional negative review.  Unfortunately I am inclined to side with the negative review and recommend that the paper not be published at this time.  While the proposed measure of sharpness appears to provide stronger correlation with the generalization performance of the overall model, the fact that basic aspects of the correlation such as the difference in the sign of the correlation that occur between experiments somewhat muddies the overall message.  This phenomenon is perhaps a more general property of sharpness in general, but I believe the work would be better served by a revision which can provide a more complete message as to the benefits and interpretation of the measure as a predictor of network generalization.  Indeed, the authors already suggest at potential directions along these lines in the discussion such as considering the properties of the data or incorporating further symmetries into the metric, and I would encourage the authors to further refine the overall implications/interpretation of the proposed metric in future versions of the manuscript.

**Additional Comments On Reviewer Discussion:**

The authors were largely responsive to the reviewers' comments, and several reviewers increased their score as a result.

---

### Decision · Program_Chairs · 2025-01-22

Reject